# STOCHASTICALLY CONTROLLED COMPOSITIONAL GRADIENT FOR COMPOSITION PROBLEMS

## ABSTRACT

We consider composition problems of the form $\frac{1}{n}\sum_{i=1}^{n} F_i(\frac{1}{n}\sum_{j=1}^{n} G_j(x))$. Composition optimization arises in many important machine learning applications: reinforcement learning, variance-aware learning, nonlinear embedding, and many others. Both gradient descent and stochastic gradient descent are straightforward solution, but both require to compute $\frac{1}{n}\sum_{j=1}^{n} G_j(x)$ in each single iteration, which is inefficient-especially when $n$ is large. Therefore, with the aim of significantly reducing the query complexity of such problems, we designed a stochastically controlled compositional gradient algorithm that incorporates two kinds of variance reduction techniques, and works in both strongly convex and non-convex settings. The strategy is also accompanied by a mini-batch version of the proposed method that improves query complexity with respect to the size of the mini-batch. Comprehensive experiments demonstrate the superiority of the proposed method over existing methods.

## 1 INTRODUCTION

In this paper, we study the following composition minimization problem,

$$\min_{x\in\mathbb{R}^N} \left\{ f(x) \stackrel{\text{def}}{=} F(G(x)) \stackrel{\text{def}}{=} \frac{1}{n}\sum_{i=1}^{n} F_i\left(\frac{1}{n}\sum_{j=1}^{n} G_j(x)\right)\right\}, \tag{1.1}$$

where $f\colon \mathbb{R}^N \to \mathbb{R}$ is differentiable and possibly non-convex, each $F_i\colon \mathbb{R}^M \to \mathbb{R}$ is a smooth function, each $G_i\colon \mathbb{R}^N \to \mathbb{R}^M$ is a mapping function, both the numbers of $F_i$'s and $G_j$'s are assumed to be $n$ for simplicity We call $G(x):= \frac{1}{n}\sum_{j=1}^{n} G_j(x)$ the inner function, and $F(w):= \frac{1}{n}\sum_{i=1}^{n} F_i(w)$ the outer function. Many machine learning problems can be cast as composition problems that include two finite-sum structures: reinforcement learning (Sutton et al., 1998; Wang et al., 2017; Liu et al., 2016), variance-averse learning (Lian et al., 2017), and nonlinear embedding (Hinton & Roweis, 2003; Dikmen et al., 2015). In particular,

- (**reinforcement learning**) The $\mathcal{S} \times \mathcal{S}$ system of Bellman equations Wang et al. (2017) can be written as $\min_{x\in\mathbb{R}^s} \|\mathbb{E}[B]x - \mathbb{E}[b]\|^2$, where $\mathbb{E}[B] = I - \gamma P^\pi$, $\gamma \in (0,1)$ is a discount factor, $P^\pi$ is the transition probability under policy $\pi$, and $\mathbb{E}[b]$ is the expected state transition reward. This is one of key problems in reinforcement learning for evaluating the value of a policy $\pi$.

- (**risk-averse learning**) The risk-averse learning Lian et al. (2017) aims to maximize the expected return while control the variance (or risk) in the meantime:
$$\min_x -\mathbb{E}_a[h(x;a)] + \lambda\text{Var}_a[h(x;a)],$$
where $h(x;a)$ is the loss function including a random variable $a$, $\lambda > 0$ is a regularization parameter.

- (**nonlinear embedding**) Stochastic nonlinear embedding Hinton & Roweis (2003) aims to map a group of points from a high dimensional space to a low dimensional space by minimizing the KL divergence. It is a non-convex composition optimization
$$\min_x \sum_t \text{KL}(p_{\cdot|t} \| q_{\cdot|t}) := \sum_t \sum_i p_{i|t} \log \frac{p_{i|t}}{q_{i|t}}, \tag{1.2}$$
where $p_{i|t}$ and $q_{i|t}$ are the conditional probabilities w.r.p. $\{z_i\}_{i=1}^n$ and $\{x_i\}_{i=1}^n$,

$$p_{i|t} = \frac{d(z_t, z_i)}{\sum_{j \neq t} d(z_t, z_j)}, q_{i|t} = \frac{d(x_t, x_i)}{\sum_{j \neq t} d(x_t, x_j)},$$

where $d(\cdot, \cdot)$ is the dissimilar distance function between two samples.

To solve the composition optimization including the finite-sum structure in (1.1), two most straight-forward approaches are the gradient descent (GD) and the stochastic gradient descent (SGD). However, it is extremely expensive to scan all the inner functions (for both SGD and GD) as well as all the outer functions (for GD) in each iteration. However, note that, unlike solving common stochastic optimization problems, randomly sampling one inner function and one outer function does not give an unbiased estimate for the true gradient; that is, $\mathbb{E}_{i \sim [n], j \sim [n]}[(\partial G_j(x))^\mathsf{T} \partial F_i(\tilde{G}(x))] \neq \nabla f(x)$, where $\tilde{G}(x)$ is the estimation of $G(x)$. The key to solving this composition objective is how to estimate the value of $G(x_k)$ and its Jacobian with high accuracy using only a few samples in each iteration.

Recently, many stochastic optimization methods solving the composition problem have been developed, such as the stochastic gradient based method and the variance-reduction based method. For example, stochastic compositional gradient descent (SCGD) (Wang et al., 2017; Liu et al., 2016) estimates the inner function $G(x)$ by using an iterative weighted average of the past values of $G(x)$, and then performs the stochastic quasi-gradient iteration. The advantage of this method is that convergence rate does not depend on $n$; however, it queries more samples to the desired point. Another set of approaches is based on variance reduction – for instance, compositional stochastic variance reduction gradient (Compositional-SVRG) (Lian et al., 2017) estimates the inner function $G(x)$ and the gradient of function $f(x)$ by using the variance reduction technique; however, the derived linear convergence rate is related to $n$. Motivated by a few recent works (Lei & Jordan, 2017; Lei et al., 2017; Allen-Zhu, 2017) that focus on the stochastically controlled gradient, we were inspired to look for a way to improve the query complexity and reduce the dependence on $n$ to solve the composition optimization in (1.1).

Hence, this paper presents a novel and more efficient method named stochastically controlled compositional gradient (SCCG) for solving composition problems involving a two-finite-sum structure. The result is improved query complexity over existing approaches. Further, all results in this paper can be easily extended to cases where the number of $F_i$ and the number of $G_j$ are different. The main contributions of this article are summarized below.

- We provide a stochastically controlled function to estimate the inner function $G(x)$. Inspired by stochastically controlled stochastic gradient (SCSG) (Lei & Jordan, 2017) that estimates the gradient, $G(x)$ can also be estimated by using a snapshot $\tilde{x}_s$, in which $G(\tilde{x}_s)$ is not computed directly, but is estimated through a random subset from $[n]$. This is the first time that a stochastically controlled function has been incorporated into the process of estimating the inner function. We have also analyzed how the size of the subset might influence the query complexity for both strongly convex and non-convex functions.

- We provide a stochastically controlled compositional gradient to estimate the $\nabla f(x)$. However, there are two potential situations that could be encountered in the estimation process that can impede convergence. First, the expectation of the gradient is no longer an unbiased estimation; and, second, the gradient of $f(\tilde{x}_s)$ at the snapshot is formed by two random subsets, which are used for the functions $F_i$ and $G_j$ respectively. Moreover, the biased gradient bring more difficulty in proving the convergence, which are greatly different from those encountered in (Lei & Jordan, 2017; Lian et al., 2017; Lei et al., 2017). To address these scenarios, we have identified a bound on the size of the subsets that are used to estimate the gradient. The details of the analysis can be referred to Section 3.1 and 3.2.

- A mini-batch version of the proposed algorithm is also provided for both strongly convex and non-convex functions. The corresponding query complexities are improved according to the size of the mini-batch. More information can be referred to Section 3.3.

## 1.1 RESULTS

Following the classical benchmark for a general problem, the composition algorithm is also to find a point x satisfying $f(x) - f(x^*) \leq \epsilon$ for a convex function, where $x^*$ is the optimal point in the strongly convex function, and $\|\nabla f(x)\|^2 \leq \epsilon$ for a non-convex function, respectively. The elegance

| Algorithm | Strongly Convex | Non-convex |
|---|---|---|
| SCGD (Wang et al., 2017) | $\mathcal{O}(1/\epsilon^{3/2})$ | $\mathcal{O}(1/\epsilon^4)$ |
| Acc-SCGD (Wang et al., 2017) | $\mathcal{O}(1/\epsilon^{5/4})$ | $\mathcal{O}(1/\epsilon^{7/2})$ |
| ASC-PG (Liu et al., 2016) | $\mathcal{O}(1/\epsilon^{5/4})$ | $\mathcal{O}(1/\epsilon^{9/4})$ |
| SC-SVRG (Liu et al., 2017b) | $\mathcal{O}\left(\left(n + L_f^2/\mu^4\right)\log\left(1/\epsilon\right)\right)$ | $\mathcal{O}(n^{4/5}/\epsilon)$ |
| mini-batch VRSC-PG (Huo et al., 2018) | $\mathcal{O}\left(\left(n + L_f^2/\mu^3\right)\log\left(1/\epsilon\right)\right)$ | $\mathcal{O}(n^{2/3}/\epsilon)$ |
| mini-batch C-SAGA (Yuan et al., 2019) | $\mathcal{O}\left(\left(n + L_f^2/\mu^3\right)\log\left(1/\epsilon\right)\right)$ | $\mathcal{O}(n^{2/3}/\epsilon)$ |
| SCCG | $\mathcal{O}\left(\left(\min\left\{n, \frac{1}{\epsilon\mu^2}\right\} + \frac{L_f^2}{\mu^2}\min\left\{n, \frac{1}{\mu^2}\right\}\right)\log\left(1/\epsilon\right)\right)$ | $\mathcal{O}(\min\{\frac{1}{\epsilon^{9/5}}, \frac{n^{4/5}}{\epsilon}\})$ |
| mini-batch SCCG (b=$\frac{1}{\mu}$,min $\left\{n, \frac{1}{\epsilon}\right\}^{\frac{2}{3}}$) | $\mathcal{O}\left(\left(\min\left\{n, \frac{1}{\epsilon\mu^2}\right\} + \frac{L_f^2}{\mu}\min\left\{n, \frac{1}{\mu^2}\right\}\right)\log\left(1/\epsilon\right)\right)$ | $\mathcal{O}(\min\{\frac{1}{\epsilon^{5/3}}, \frac{n^{2/3}}{\epsilon}\})$ |

Table 1: Comparison of the query complexity with different algorithms. Note: $\mu$ and $L_f$ are defined in the Preliminary Section. $b$ is the size of the mini-batch.

of a composition algorithm is evaluated based on its query complexity, defined as the number of queries in a given sampling oracle that are needed to compute the gradient. Here, we give the query complexities of the composition problem in Table 1, which offers an insightful comparison to other algorithms.

**Strongly convex function** The query complexity for the strongly convex function is $\mathcal{O}((\min\{n, 1/(\epsilon\mu^2)\} + L_f^2/\mu^2\min\{n, 1/\mu^2\})\log(1/\epsilon))$. The result is the general form for the strongly convex composition and is equal to or better than the query complexity in (Lian et al., 2017) and (Liu et al., 2017a).

**Non-convex function** The query complexity is $\mathcal{O}(\min\{1/\epsilon^{9/5}, n^{4/5}/\epsilon\})$, which is better than the result in (Liu et al., 2016) and comparable to the result in (Liu et al., 2017b).

**Mini-batch** [1] For the mini-batch version, the query complexity can be improved to some extent, that is $\mathcal{O}((\min\{n, 1/(\epsilon\mu^2)\} + L_f^2/(b\mu^2)\min\{n, 1/\mu^2\})\log(1/\epsilon))$. and $\mathcal{O}(\min\{1/\epsilon^{9/5}, n^{4/5}/\epsilon\}/b^{1/5})$ for strongly convex and non-convex functions, respectively, which are better than mini-batch variance reduced stochastic compositional proximal gradient method (VRSC-PG) (Huo et al., 2018) and mini-batch Composite SAGA (C-SAGA) (Yuan et al., 2019) when b=$1/\mu$ and min $\{n, 1/\epsilon\}^{2/3}$ for strongly convex and non-convex functions, respectively.

## 1.2 RELATED WORK

As the amount of data we have at our disposal grows, stochastic optimization has become a popular technique in the realm of machine and deep learning, particularly for optimizing finite-sum functions. The typical algorithms for solving such problems include stochastic gradient descent (Ghadimi & Lan, 2016), SVRG (Johnson & Zhang, 2013; Reddi et al., 2016), stochastic dual coordinate ascent (SDCA) (Shalev-Shwartz & Zhang, 2014; 2013) and the accelerated Nesterov's method (Nesterov, 2013), accelerated randomized proximal coordinate (APCG) (Lin et al., 2014; 2015) and Katyusha method (Allen-Zhu, 2017). The standard procedure for optimizing a problem with a finite-sum structure is to randomly select one or a block of components to estimate the gradient. However, knowing that the estimated gradient usually has a large variance, the gradient of the function is estimated from a snapshot to appropriately reduce the variance – in other words, the procedure includes a variance reduction mechanism.

Composition optimization problems can also be solved with the above algorithms, but the two-finite-sum structures in composition problems mean that when the gradient of the inner function is estimated directly, the query complexity can substantially increase. Recently, Wang et al. (2017) proposed a method based on first-order SCGD to overcome this issue where the variable and the inner function are updated alternately in two steps. The method has a query complexity of $\mathcal{O}(\epsilon^{-7/2})$ for a general function and $\mathcal{O}(\epsilon^{-5/4})$ for a strongly convex function. Liu et al. (2016) employed Nes-

---

[1]$b$ denotes the size of the mini-batch, can be obtained through the $\eta \leq 1$ from Theorem 1 and Theorem 2.

terov's method to accelerate the composition problem, reaching $\mathcal{O}(\epsilon^{-5/4})$ and $\mathcal{O}(\epsilon^{-9/4})$ for strongly convex and non-convex functions, respectively. Ghadimi et al. (2018) proposed a nested averaged stochastic approximation method to find an approximate stationary point within the problem, resulting in a sample complexity of $\mathcal{O}(1/\epsilon^2)$. However, these methods estimate the inner function using an iterative weighted average of the past function.

The other stream of solutions focuses on variance reduction technology. For instance, Lian et al. (2017) initially applied the SVRG-based method to estimate the inner function $G(x)$ and the gradient of the function $f(x)$, which yields a linear convergence rate. Subsequently, Liu et al. (2017a) and Devraj & Chen (2019) applied a dual-based method to composition problem, which also yields a linear convergence rate. Devraj & Chen (2019) also applied the stochastic variance reduced primal-dual algorithms to composition problem. Yu & Huang (2017) turned to an ADMM-based Boyd (2011) method and provided an analysis of convex functions that do not rely on Lipschitz smoothness. Moreover, Liu et al. (2017b) went a step further and considered non-convex functions, analyzing the query complexity with both inner and outer functions of different sizes. Lin et al. (2018) considered non-smooth convex composition functions, offering an incremental first-order oracle complexity analysis. Zhang & Xiao (2019) and Huo et al. (2018) also provided an randomized incremental gradient method for the composition problem including regularization.

Many recent articles have discussed variance reduction methods that estimate the gradient from a random subset rather than through direct computation. Lei & Jordan (2017), for example, proposed an SCSG method for a convex finite-sum function. They then applied it to a non-convex problem in (Lei et al., 2017) by using less than a single pass to compute the gradient at the snapshot point. Furthermore, Allen-Zhu (2017) proposed the Natasha1.5 algorithm, in which the gradient for each epoch is based on a random subset. Moreover, the objective function has the regularization term. Liu et al. (2018) applied an SCSG based method to the zeroth-order optimization problems with the finite-sum function. Recently, Yuan et al. (2019) applied the stochastic recursive gradient descent method to the composition problem.

The rest of paper is organized as follows: in Section 2, we give preliminaries used for analyzing the proposed algorithm. Section 3 presents the SCSG-based method for the strongly convex and non-convex composition problem and the corresponding mini-batch version. In Section 4, we give the experimental results. We conclude our paper in Section 5.

## 2 PRELIMINARIES

Throughout this paper, we use the Euclidean norm denoted by $\|\cdot\|$. We use $i \in [n]$ to denote that $i$ is generated from $[n] = \{1, 2, ..., n\}$. We denote by $(\partial G(x))^{\mathsf{T}} \nabla F(G(x))$ the full gradient of the function $f$, $\partial G(x)$ the Jacobian of $G$, and $(\partial G_j(x))^{\mathsf{T}} \nabla F_i(G(x))$ as the stochastic gradient of the function $f$, where $i$ and $j$ are randomly and independently selected from $[n]$. We use $A = |\mathcal{A}|$ to denote the number of elements in the set $\mathcal{A}$, and define $G_{\mathcal{A}}(x) = \frac{1}{A} \sum_{1 \le j \le A} G_{\mathcal{A}[j]}(x)$. We use $\mathbb{E}$ to denote the expectation, that is $\mathbb{E}_{\mathcal{A}}[v] = \frac{1}{A} \sum_{1 \le i \le A} v_{\mathcal{A}[i]}$. Note that all the variables such as subsets $\mathcal{A}$ and $\mathcal{B}$, elements $i$ and $j$ are independently selected from $[n]$, in particular, the element in $\mathcal{A}$ and $\mathcal{B}$ are independent. So we use $\mathbb{E}$ in instead of $\mathbb{E}_i$, $\mathbb{E}_j$, $\mathbb{E}_{\mathcal{A}}$ and $\mathbb{E}_{\mathcal{B}}$ except when explicitly stated otherwise. Recall definitions on Lipschitz function, smooth function and strongly convex.

**Definition 1.** *For function $p$ on $\mathcal{X}$, $\forall x, y \in \mathcal{X}$, A function $p$ is a $B_p$-Lipschitz, that is $\|p(x) - p(y)\| \le B_p \|x - y\|$; A function $p$ is a $L_p$-smooth, that is $\|\nabla p(x) - \nabla p(y)\| \le L_p \|x - y\|$; A function $p$ is a $\mu$-strongly convex, that is $p(y) \ge p(x) + \langle p(x), y - x \rangle + \mu/2 \|x - y\|^2$.*

Through our discussions, we make the following assumptions,

**Assumption 1.** *Let $B_G$, $L_F$ and $L_f$ be positive scalars,* [2]

- $G_j$ *is $B_G$-Lipschitz, $j \in [n]$, that is $\|G_j(x) - G_j(y)\| \le B_G \|x - y\|$.*

- $F_i$ *is $L_F$-smooth, $i \in [n]$, that is $\|\nabla F_i(x) - \nabla F_i(y)\| \le L_F \|x - y\|$.*

---

[2]In the strongly convex composition problem, the upper bounded Jacobian does not imply that the gradient of $f(x)$ is upper bounded since we do not require the gradient of $F_i$ is upper bounded. Moreover, in the experimental section, we will show that the Jacobian of G(x) is bounded.

- *For function $F_i(G(x))$, there exists a constant $L_f$ satisfying $\|(\partial G_j(x))^\mathsf{T}\nabla F_i(G(x)) - (\partial G_j(y))^\mathsf{T}\nabla F_i(G(y))\| \le L_f\|x - y\|, \forall i, j \in [n]$.*

- *We assume that $i$ and $j$ are independently and randomly selected from $[n]$, $z \in \mathbb{R}^M$, $x \in \mathbb{R}^N$, then $\mathbb{E}[(\partial G_j(x))^\mathsf{T}\nabla F_i(z)] = (\partial G(x))^\mathsf{T}\nabla F(z)$.*

Furthermore, we define $H_1$ and $H_2$ are the upper bounds on the variance of $G(x)$ and $(\partial G(x))^\mathsf{T}\nabla F(y)$, respectively, that is,

$$\frac{1}{n}\sum_{i=1}^{n}\|G(x) - G_i(x)\|^2 \le H_1, \frac{1}{n^2}\sum_{j=1}^{n}\sum_{i=1}^{n}\left\|(\partial G(x))^\mathsf{T}\nabla F(y) - (\partial G_j(x))^\mathsf{T}\nabla F_i(y)\right\|^2 \le H_2.$$

In the paper, we denote by $x_k^s$ the $k$-th inner iteration at $s$-th epoch. But in each epoch analysis, we drop the superscript $s$ and denote by $x_k$ for $x_k^s$. We let $x^*$ be the optimal solution of the convex $f(x)$. Throughout the convergence analysis, we use $\mathcal{O}(\cdot)$ notation to avoid many constants, such as $B_G$, $L_F$, and $L_f$, that are irrelevant with the convergence rate.

# 3 STOCHASTICALLY CONTROLLED COMPOSITIONAL GRADIENT

In this section, we present the variance-reduction based method for the composition problem, which can be used for both the strongly convex function and non-convex function. Before describing the proposed algorithm, we recall the original SVRG (Johnson & Zhang, 2013). The general process of SVRG works as follows. The update process is divided into $S$ epochs, and each of the epoch consists of $K$ iterations. At the beginning of each epoch, SVRG defines a snapshot vector $\tilde{x}_s$, and then compute the full gradient $\nabla f(\tilde{x}_s)$. In the inner iteration of the current epoch, SVRG defines the estimated gradient by randomly selecting $i_k$ from [n] at the $k$-th iteration,

$$(\partial G(x_k))^\mathsf{T}\nabla F_{i_k}(G(x_k)) - (\partial G(\tilde{x}_s))^\mathsf{T}\nabla F_{i_k}(G(\tilde{x}_s)) + \nabla f(\tilde{x}_s). \tag{3.1}$$

However, for the composition problem, there are also variance-reduction based methods in (Lian et al., 2017; Liu et al., 2017a;b). The difference with SVRG is that there is another estimated function for $G(x)$, which also has the finite-sum structure. These methods define the estimated function as

$$\tilde{G}_k = G_{\mathcal{A}}(x_k) - G_{\mathcal{A}}(\tilde{x}_s) + G(\tilde{x}_s), \tag{3.2}$$

where $\mathcal{A}$ is the mini-batch formed by randomly sampling from $[n]$. Whereas, as the number of the inner function $G_j$ and the outer function $F_i$ increase, it is not reasonable to compute the full gradient of $f(x)$ and the full function $G(x)$ directly for each epoch.

Extending from the SCSG (Lei et al., 2017; Lei & Jordan, 2017) and Natasha1.5 (Allen-Zhu, 2017), we present a new algorithm SCCG for the composition problem as shown in Algorithm 1. [3] We introduce two subsets $\mathcal{D}_1$ and $\mathcal{D}_2$, which are independent with each other and randomly selected from $[n]$, respectively. We define $\mathcal{D} = \mathcal{D}_1 \cup \mathcal{D}_2$ as a new variable, which is important in analyzing the convergence. Firstly, $\mathcal{D}_1$ is used for estimating the inner function. Based on the variance reduction technique, the estimated inner function at $k$-th iteration of $s$-th epoch is

$$\hat{G}_k = G_{\mathcal{A}}(x_k) - G_{\mathcal{A}}(\tilde{x}_s) + G_{\mathcal{D}_1}(\tilde{x}_s), \tag{3.3}$$

where the subset of $\mathcal{A}$ is the same as in (3.2). Note that $\mathcal{A}$ and $\mathcal{D}$ are independent with each other. The difference with (3.2) is that the third term in (3.3) is computed under the subset $\mathcal{D}_1$ rather than $[n]$. Throughout the paper, we assume that $|\mathcal{A}| \le |\mathcal{D}_1|$. Secondly, $\mathcal{D}_2$ is used to estimate the outer function $F$. The key distinguish with (Lei et al., 2017; Lei & Jordan, 2017; Allen-Zhu, 2017) is the biased full gradient of $f(\tilde{x}_s)$. We define this estimated full gradient of $f(\tilde{x}_s)$ for each epoch as $\nabla\hat{f}_{\mathcal{D}}(\tilde{x}_s) = (\partial G_{\mathcal{D}_1}(\tilde{x}_s))^\mathsf{T}\nabla F_{\mathcal{D}_2}(G_{\mathcal{D}_1}(\tilde{x}_s))$. Though $\mathbb{E}_{\mathcal{A},\mathcal{D}}[\nabla\hat{f}_{\mathcal{D}}(\tilde{x}_s)] \ne \nabla f(\tilde{x}_s)$, we could still estimate the gradient of the $f(x_k)$ by

$$\nabla\tilde{f}_k = (\partial G_{j_k}(x_k))^\mathsf{T}\nabla F_{i_k}(\hat{G}_k) - (\partial G_{j_k}(\tilde{x}_s))^\mathsf{T}\nabla F_{i_k}(G_{\mathcal{D}_1}(\tilde{x}_s)) + \nabla\hat{f}_{\mathcal{D}}(\tilde{x}_s), \tag{3.4}$$

---

[3]The parameters' setting can be referred to Theorem 1 and Theorem 2 for the strongly convex and non-convex function, respectively.

---

**Algorithm 1** Stochastically Controlled Compositional Gradient (SCCG) for the strongly convex or non-convex composition problem

---

**Require:** $K$, $S$, $\eta$ , $\tilde{x}_0$ and $\mathcal{D} = \mathcal{D}_1 \cup \mathcal{D}_2$, where $\mathcal{D}_1$ and $\mathcal{D}_2$ are mini-batches.
   **for** $s = 0, 1, 2, \cdots, S-1$ **do**
       Sample from $[n]$ for D times to form mini-batch $\mathcal{D}_1$
       Sample from $[n]$ for D times to form mini-batch $\mathcal{D}_2$
       $\nabla \hat{f}_{\mathcal{D}}(\tilde{x}_s) = (\partial G_{\mathcal{D}_1}(\tilde{x}_s))^\mathsf{T} \nabla F_{\mathcal{D}_2}(G_{\mathcal{D}_1}(\tilde{x}_s))$
       $x_0 = \tilde{x}_s$
       **for** $k = 0, 1, 2, \cdots, K-1$ **do**
          Sample from $[n]$ to form mini-batch $\mathcal{A}$
          $\hat{G}_k = G_{\mathcal{A}}(x_k) - G_{\mathcal{A}}(\tilde{x}_s) + G_{\mathcal{D}_1}(\tilde{x}_s)$
          Uniformly and randomly pick $i_k$ and $j_k$ from $[n]$
          Compute the estimated gradient $\nabla \tilde{f}_k$ from (3.4)
          $x_{k+1} = x_k - \eta \nabla \tilde{f}_k$
       **end for**
       Update $\tilde{x}_{s+1} = x_K$, or $\tilde{x}_{s+1} = x_r$, $r$ is randomly selected from $[K-1]$
   **end for**
   Output: $\hat{x}_k^s$ is uniformly and randomly chosen from $s \in \{0, ..., S-1\}$ and $k \in \{0, .., K-1\}$.

---

where $i_k$ and $j_k$ are randomly selected from [n] at the $k$-th iteration for functions $F$ and $G$, respectively. Furthermore, $\mathbb{E}_{i_j, j_k \mathcal{A}, \mathcal{D}}[\nabla \tilde{f}_k] \neq \nabla f(x_k)$ as well. This gives us more discussion about the upper bound with respect to the estimated function and the gradient under the new random subset $\mathcal{D}$ (more details can be referred to appendix).

### 3.1 SCCG FOR THE STRONGLY CONVEX CASE

In this subsection, we analyze the query complexity for the strongly convex composition problem and show that our result is better or comparable to the previous methods. Furthermore, we discuss the query complexity under different value with respect to $n$, $\mu$ and $\epsilon$

**Theorem 1.** *In Algorithm 1, for the $\mu$-strongly convex problem, suppose Assumption 1 holds, let the step size is $\eta \leq \mu/(135 L_f^2)$, the subset size of $\mathcal{A}$ is $A = \min\{n, 128 B_G^4 L_F^2/\mu^2\}$, the subset size of $\mathcal{D}_1$ and $\mathcal{D}_2$ are both $D = \min \left\{ n, 5 \left(16 B_G^4 L_F^2 H_1 + 4 H_2\right)/(4\epsilon\mu^2) \right\}$, the number of the inner iteration is $K \geq 540 L_f^2/\mu^2$, the number of outer iteration is $S \geq 1/(\log(1/\rho)) \log(2E\|\tilde{x}_0 - x^*\|^2/\epsilon)$.*

*Then, the query complexity is $(D + KA) S = \mathcal{O}\left(\left(\min\left\{n, \frac{1}{\epsilon\mu^2}\right\} + \frac{L_f^2}{\mu^2} \min\left\{n, \frac{1}{\mu^2}\right\}\right) \log(1/\epsilon)\right).$*

As can be seen from the above result, Theorem 1 presents the general query complexity under different parameters ( the details of parameters' setting can be referred to the Appendix.). Comparing $n$ with corresponding parameters, we analyze the query complexity separately. We remove the parameters such as $B_G^2$, $L_F^2$, $H_1$ and $H_2$, and analyze the size with the order of $1/\mu^2$. We consider three internals of $n$ while the min value of the function in the above query complexity will take different results:

- $1/\mu^2 \leq 1/(\epsilon\mu^2) \leq n$. When $n$ is large enough such that we can obtain the query complexity is $\mathcal{O}((1/(\epsilon\mu^2) + L_f^2/\mu^4) \log(1/\epsilon))$. This result avoids the situation that computing the full gradient of $f(x)$ and the full function $G(x)$ for the large-scale number of $n$. What's more, this result is better than Compositional-SVRG (Lian et al., 2017; Liu et al., 2017a).

- $1/\mu^2 \leq n \leq 1/(\epsilon\mu^2)$. When $n$ is smaller than $1/(\epsilon\mu^2)$, the query complexity becomes $\mathcal{O}((n + L_f^2/\mu^4) \log(1/\epsilon))$, which is the same as Compositional-SVRG (Lian et al., 2017). However, we need to compute the full gradient of $\nabla f(\tilde{x}_s)$ as in (3.1). The estimation of inner function $G(x)$ is the same as in (Lian et al., 2017).

- $n \leq 1/\mu^2 \leq 1/(\epsilon\mu^2)$. When $n$ is the smallest one, the query complexity becomes $\mathcal{O}((n + L_f^2 n/\mu^2) \log(1/\epsilon))$. The result has a similar form to SVRG (Johnson & Zhang, 2013). This also gives us an intuition that the inner function should be computed directly rather than estimated if $n$ is small.

### 3.2 SCCG FOR THE NONCONVEX CASE

In the previous subsection, we showed convex SCCG converges to the optimal point with improved query complexity. A natural question is whether the proposed algorithm can improve the performance of the non-convex problem. We provide an affirmative answer. In this subsection, we present the query complexity for the non-convex composition problem in term of stationarity gap $||\nabla f(x)||^2$.

**Theorem 2.** *In Algorithm 1, for non-convex function, suppose Assumption 1 holds, let the step size is $\eta = \min\{1/n^{2/5}, \epsilon^{2/5}\}$, the size of the subset $\mathcal{D}_1$ and $\mathcal{D}_2$ are $D = \min\{n, \mathcal{O}(1/\epsilon)\}$, the size of subset $\mathcal{A}$ is $A = \min\{n, \mathcal{O}(1/\eta)\}$, the number of inner iteration is $K \leq \mathcal{O}\left(1/\eta^{3/2}\right)$, the total number of iteration is $T = \mathcal{O}(1/(\epsilon\eta))$, in order to obtain $\mathbb{E}[||\nabla f(\hat{x}_k^s)||^2] \leq \epsilon$, the query complexity is $\mathcal{O}\left(\min\left\{1/\epsilon^{9/5}, n^{4/5}/\epsilon\right\}\right)$.*

From the above result, we analyze the query complexity of the non-convex problem separately: 1) when $n \geq 1/\epsilon$, our query complexity becomes $\mathcal{O}(1/\epsilon^{9/5})$, which is independent on $n$. This is better than the query complexity of the accelerated method in (Liu et al., 2016), in which the query complexity does not depend on $n$ as well. 2) when $n \leq 1/\epsilon$, the query complexity becomes $\mathcal{O}(n^{4/5}/\epsilon)$, which is consistent with the result of (Liu et al., 2017b) in solving the problem (1.1).

### 3.3 MINI-BATCH VERSION OF SCCG

In this subsection, we present the mini-batch version of the proposed method in Algorithm 2 (in appendix) and obtain the corresponding query complexities for both the strongly convex and the non-convex functions, which provably benefit from mini-batching. As the process of the proof is similar to that of Theorem 1 and Theorem 2, and the difference with Algorithm 1 is the computation of the gradient of $f(x)$ (the corresponding proof of bound is in appendix), we could directly present the corresponding results for both the strongly convex and the non-convex problems.

**Corollary 1.** *In Algorithm 2, for the $\mu$-strongly convex problem, suppose Assumption 1 holds, let the step size $\eta \leq b\mu/(135L_f^2)$, the number of the inner iteration is $K \geq 540L_f^2/(b\mu^2)$, in order to obtain $\mathbb{E}||\tilde{x}_s - x^*||^2 \leq \epsilon$, the query complexity is $\mathcal{O}\left(\left(\min\left\{n, \frac{1}{\epsilon\mu^2}\right\} + \frac{L_f^2}{b\mu^2}\min\left\{n, \frac{1}{\mu^2}\right\}\right)\log(1/\epsilon)\right)$.*

**Corollary 2.** *In Algorithm 2, for the non-convex problem, suppose Assumption 1 holds, let the step size $\eta = b^{3/5}\min\{1/n^{2/5}, \epsilon^{2/5}\}$, the number of the inner iteration is $K \leq \mathcal{O}\left(b^{1/2}/(\eta^{3/2})\right)$, in order to obtain $\mathbb{E}[||\nabla f(\hat{x}_k^s)||^2] \leq \epsilon$, the query complexity is $(1/b^{1/5})\mathcal{O}\left(\min\left\{1/\epsilon^{9/5}, n^{4/5}/\epsilon\right\}\right)$.*

From the above-given query complexity results for the strongly convex and non-convex problems, we can see that both of their step size $\eta$ and the number of inner iteration $K$ are larger than the corresponding ones in the non-mini-batch version. These two key parameters lead to the improved query complexity for both strongly convex and non-convex functions.

## 4 EXPERIMENTS

In this section, we evaluate the performance of our proposed algorithm on the strongly convex and non-convex functions, respectively.

**SCCG for strongly convex function**[4] To verify the effectiveness of the algorithm, we use the mean-variance optimization in portfolio management[5]:

$$\min_{x \in \mathbb{R}^d} -\frac{1}{n}\sum_{i=1}^n \langle r_i, x \rangle + \frac{1}{n}\sum_{i=1}^n (\langle r_i, x \rangle - \frac{1}{n}\sum_{i=1}^n \langle r_i, x \rangle)^2,$$

where $r_i \in \mathbb{R}^N, i \in [n]$ is the reward vector, and $x \in \mathbb{R}^N$ is the invested quantity. In the experimental setting, we set $n$=3000, $|\mathcal{A}| \approx n^{2/3}$, $|\mathcal{D}_1| = 2400, 2600, 2800$, which are denoted as SCCG (2400), SCCG (2600) and SCCG (2800). The reward vectors are generated on Gaussian distribution

---

[4]Our aim is to compare our general variance-reduce based method with the stochastic composition gradient method, and also to verify the proposed algorithm, thus we do not include SVRG-based method.

[5]This formulation is just used to verify our proposed algorithm. In appendix, we show the bounded Jacobian.

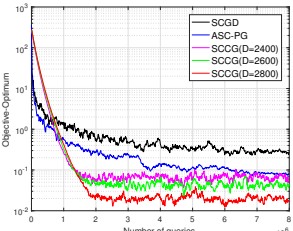 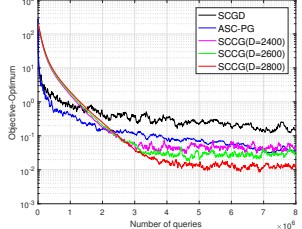 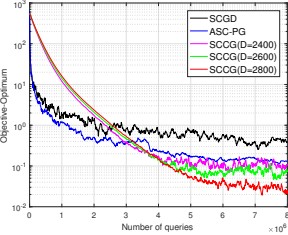

Figure 1: Strongly convex: Comparison of the gap between the function value and the optimal value among SCGD, ASC-PG and SCCG methods. Dataset (from left to right): condition numbers of the covariance matrix are set $\kappa_{cov}$ =10, 30 and 50, respectively.

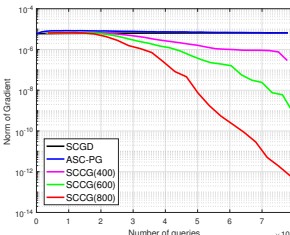 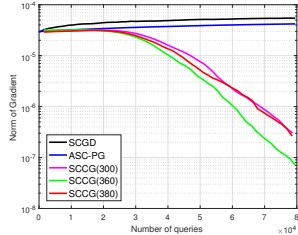 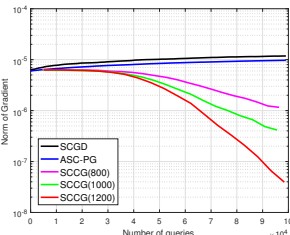

Figure 2: Non-convex: Comparison of the norm of the gradient between SCGD, ASC-PG and SCCG; Dataset (from left to right): mnist, olivettifaces and coil20.

with the condition number of its covariance matrix denoted by $\kappa_{cov}$. Furthermore, we consider three conditions numbers, $\kappa_{cov}$=10, 30 and 50. We compare our algorithm with the stochastic gradient based methods SCGD and accelerated stochastic method ASC-PG. Figure 1 shows the performance of the gap between the value function and optimal value, we observe that our algorithm is better than stochastic gradient methods, SCGD and ASC-PG.

**SCCG for non-convex function** For the non-convex function, we apply the proposed SCCG method to the nonlinear embedding problem in (1.2). We consider the distance of low-dimension space between $x_i$ and $x_j$ as $1/(1 + \|x_i - x_j\|^2)$, $i, j \in [n]$. Then, the problem can be formulated as the problem in (1.1), in which the details can be referred to the appendix. We consider three datasets: mnist, Olivetti faces and COIL-20 including different sample sizes and dimensions, $1000\times$ 784, $400 \times 4096$ and $1440\times$ 16384. Our experiment is to verify our proposed algorithm, thus we set $\mathcal{D}_1 = \mathcal{D}_2$ in default and choose three different sizes of sample set $\mathcal{D}_1$, which are smaller than $n$. For example, for the case of mnist, we choose $|\mathcal{D}_1| = 400, 600, 800$, which are denoted as SCCG (400), SCCG (600) and SCCG (800). Furthermore, we also set $|\mathcal{A}|\approx n^{2/3}$, where $n$ is the total number of samples. Figure 2 shows the norm of the gradient, and Figure 3 (in appendix) shows the objection value. We compare our algorithm with the stochastic gradient based method (SCGD and ASC-PG), and observe that our proposed algorithm is better than SCGD and ASC-PG on both the norm of the gradient and objective function. Additional experiments on reinforcement learning are given in appendix.

## 5 CONCLUSION

In this paper, we propose the variance reduction based method for the strongly convex and non-convex composition problems. We apply the stochastically controlled stochastic gradient to estimate inner function $G(x)$ and the gradient of $f(x)$. The query complexity of our proposed algorithm is better than or equal to the current methods on both strongly convex and non-convex functions. Furthermore, we also present the corresponding mini-batch version of the proposed method, in which the query complexities are improved as well. Experimental results also confirm that our algorithm achieves better query complexity in a real-world problem.

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

# A  TECHNICAL TOOL

For the subset $\mathcal{A} \subseteq [n]$, we present the following lemma that the variance of a random variable decreases by a factor $|\mathcal{A}|$ if we choose $|\mathcal{A}|$ independent elements from $[n]$ and average them. The proof process is trivial However, it is an important tool for analyzing the query complexity under the different sizes of the subsets.

**Lemma 1.** *If $v_1, ..., v_n \in \mathbb{R}^d$ satisfy $\sum_{i=1}^{n} v_i = \vec{0}$, and $\mathcal{A}$ is a non-empty, uniform random subset of $[n]$ and $A = |\mathcal{A}|$, that is elements in $\mathcal{A}$ are uniformly selected from $[n]$ without replacement, then*

$$\mathbb{E}_{\mathcal{A}} \left\| \tfrac{1}{A} \sum_{b \in \mathcal{A}} v_b \right\|^2 \leq \tfrac{\mathbb{I}(A < n)}{A} \tfrac{1}{n} \sum_{i=1}^{n} ||v_i||^2.$$

*Furthermore, if elements in $\mathcal{A}$ are independently selected from $[n]$ with replacement,, then*

$$\mathbb{E}_{\mathcal{A}} \left\| \tfrac{1}{A} \sum_{b \in \mathcal{A}} v_b \right\|^2 = \tfrac{1}{An} \sum_{i=1}^{n} ||v_i||^2.$$

*Proof.* Based on the $\sum_{i=1}^{n} v_i = \vec{0}$, and permutation and combination,

For the case that $\mathcal{A}$ is a non-empty, uniformly random subset of $[n]$, we have

$$\begin{aligned}
\mathbb{E}_{\mathcal{A}} \left\| \sum_{b \in \mathcal{A}} v_b \right\|^2 &= \mathbb{E}_{\mathcal{A}} \left[ \sum_{b \in \mathcal{A}} \|v_b\|^2 \right] + \frac{1}{C_n^A} \sum_{i \in [n]} \left\langle v_i, \frac{C_{n-1}^{A-1}(A-1)}{n-1} \sum_{i \neq j} v_j \right\rangle \\
&= A \frac{1}{n} \sum_{i=1}^{n} \|v_i\|^2 + \frac{A(A-1)}{n(n-1)} \sum_{i \in [n]} \left\langle v_i, \sum_{i \neq j} v_j \right\rangle \\
&= A \frac{1}{n} \sum_{i=1}^{n} \|v_i\|^2 + \frac{A(A-1)}{n(n-1)} \sum_{i \in [n]} \langle v_i, -v_i \rangle \\
&= \frac{A(n-A)}{(n-1)} \frac{1}{n} \sum_{i=1}^{n} \|v_i\|^2 \\
&\leq A \mathbb{I}(A < n) \frac{1}{n} \sum_{i=1}^{n} \|v_i\|^2,
\end{aligned}$$

where $C_A^n$ refer to the number of the combination of n things taken A at a time without repetition. Thus, we have

$$\mathbb{E}_{\mathcal{A}} \left\| \frac{1}{A} \sum_{b \in \mathcal{A}} v_b \right\|^2 = \frac{1}{A^2} \mathbb{E}_{\mathcal{A}} \left\| \sum_{b \in \mathcal{A}} v_b \right\|^2 \leq \frac{\mathbb{I}(A < n)}{A} \frac{1}{n} \sum_{i=1}^{n} \|v_i\|^2.$$

For the case that the element in $\mathcal{A}$ is randomly and independently selected from $[n]$, we have

$$\begin{aligned}
\mathbb{E}_{\mathcal{A}} \left\| \sum_{b \in \mathcal{A}} v_b \right\|^2 &= \mathbb{E}_{\mathcal{A}} \left[ \sum_{b \in \mathcal{A}} \|v_b\|^2 \right] + 2 \mathbb{E}_{\mathcal{A}} \left[ \sum_{1 \leq b < A} \left\langle v_b, \sum_{b < k \leq A} v_k \right\rangle \right] \\
&= B \frac{1}{n} \sum_{i=1}^{n} \|v_i\|^2 + 2 \mathbb{E}_{\mathcal{A}} \left[ \sum_{1 \leq b < A} \left\langle \mathbb{E}[v], \sum_{b < k \leq A} v_k \right\rangle \right] \\
&= A \frac{1}{n} \sum_{i=1}^{n} \|v_i\|^2 + A(A-1) \|\mathbb{E}[v]\|^2 \qquad\qquad\qquad \text{(A.1)} \\
&= A \frac{1}{n} \sum_{i=1}^{n} \|v_i\|^2.
\end{aligned}$$

$\square$

Based on Lemma 1, we can obtain the inequality with two-variables $\mathcal{D}_1$ and $\mathcal{D}_2$, which are used for the gradient of $f(x)$.

**Lemma 2.** *If $w_1, ..., w_n \in \mathbb{R}^{M \times N}$ and $v_1, ..., v_n \in \mathbb{R}^M$ satisfy $(\frac{1}{n} \sum_{i \in [n]} w_i)^\mathsf{T} (\frac{1}{n} \sum_{j \in [n]} v_j) = \bar{w}^\mathsf{T} \bar{v}$, and $\mathcal{D} = [\mathcal{D}_1, \mathcal{D}_2]$ is a non-empty, uniform random subset consist of $\mathcal{D}_1$ and $\mathcal{D}_2$, which are independently and uniformly selected from $[n]$, $D = |\mathcal{D}_1| = |\mathcal{D}_2|$, then*

$$
\mathbb{E}_{\mathcal{D}} \left\| \frac{1}{|\mathcal{D}_1| \, |\mathcal{D}_2|} \left( \sum_{d_1 \in \mathcal{D}_1} w_{d_1} \right)^\mathsf{T} \left( \sum_{d_2 \in \mathcal{D}_2} v_{d_2} \right) - \bar{w} \bar{v} \right\|^2
$$

$$
= \mathbb{E}_{\mathcal{D}} \left\| \frac{1}{D^2} \left( \sum_{[d_1, d_2] \in \mathcal{D}} \left( (w_{d_1})^\mathsf{T} v_{d_2} - \bar{w}^\mathsf{T} \bar{v} \right) \right) \right\|^2
$$

$$
\leq \frac{\mathbb{I}\left(D^2 < n^2\right)}{D^2} \frac{1}{n^2} \sum_{i,j=1}^n \left\| (w_i)^\mathsf{T} v_j - \bar{w}^\mathsf{T} \bar{v} \right\|^2.
$$

# B   BOUND ANALYSIS OF SCCG FOR THE COMPOSITION PROBLEM

## B.1   BOUNDS ANALYSIS OF THE ESTIMATED FUNCTION AND THE GRADIENT

Here, we mainly give different kinds of bounds for the proposed algorithm, such as $\mathbb{E}_{\mathcal{A}, \mathcal{D}_1} \| \hat{G}_k - G(x_k) \|^2$, $\mathbb{E}_{\mathcal{D}} \| E_{\mathcal{A}, i_k, j_k} [\nabla \tilde{f}_k] - \nabla f(x_k) \|^2$ and $\mathbb{E}_{i_k, j_k, \mathcal{A}, \mathcal{D}} \| \nabla \tilde{f}_k - \nabla f(x_k) \|^2$. These bounds will be used to analyze the convergence rate and query complexity. These bounds are all based on Assumption 1. Parameters such as $B_G$, $B_F$, $L_G$, $L_F$ and $L_f$ in the bound are all from these Assumptions. We do not define the exact value of parameters such as $h$, $A$ and $D$, which have a great influence on the convergence but will be clearly defined in the query analysis. Our proposed bounds are similar to that of (Lian et al., 2017; Liu et al., 2017a;b), but, the difference lies on that there is an extra subset $\mathcal{D}$, which shows an interesting phenomenon. That is when the subset $\mathcal{D}$ is equal to $[n]$, the corresponding bounds are the same as in (Lian et al., 2017; Liu et al., 2017a;b). However, it is the independent subset $\mathcal{D}$ that gives more general query complexity result for the problem (1.1). The following bounds are all used for the composition problem for both convex and non-convex problems based on the Lemma 1 and Lemma 2. For simplicity, we drop the superscript $i_k$, $j_k$, $\mathcal{A}$ and $\mathcal{D}$ for the expectation with $\mathbb{E}$ in the proof.

**Lemma 3.** *Suppose Assumption 1 holds, for $\hat{G}_k$ defined in (3.3) with $D = |\mathcal{D}_1|$ and $A = |\mathcal{A}|$, we have*

$$
\mathbb{E}_{\mathcal{A}, \mathcal{D}_1} \| \hat{G}_k - G(x_k) \|^2 \leq 4 \frac{\mathbb{I}(A < n)}{A} B_G^2 \mathbb{E} \| x_k - \tilde{x}_s \|^2 + 6 \frac{\mathbb{I}(D < n)}{D} H_1.
$$

*Proof.* By the definition of $\hat{G}_k$ in (3.3), we have

$$
\mathbb{E} \| \hat{G}_k - G(x_k) \|^2 = \mathbb{E} \| \hat{G}_k - G_{\mathcal{D}_1}(x_k) + G_{\mathcal{D}_1}(x_k) - G(x_k) \|^2
$$

$$
\overset{\text{①}}{\leq} 2\mathbb{E} \| \hat{G}_k - G_{\mathcal{D}_1}(x_k) \|^2 + 2\mathbb{E} \| G_{\mathcal{D}_1}(x_k) - G(x_k) \|^2
$$

$$
\overset{\text{②}}{\leq} 4 \frac{\mathbb{I}(A < n)}{A} B_G^2 \mathbb{E} \| x_k - \tilde{x}_s \|^2 + 6 \frac{\mathbb{I}(D < n)}{D} H_1,
$$

where ① follows from $\| a_1 + a_2 \|^2 \leq 2a_1^2 + 2a_2^2$; ② is based on Lemma 1 and the following inequality: Through adding and subtracting the term $G(x_k) - G(\tilde{x}_s)$, we have

$$
\mathbb{E} \| \hat{G}_k - G_{\mathcal{D}_1}(x_k) \|^2
$$

$$
= \mathbb{E} \| G_{\mathcal{A}}(x_k) - G_{\mathcal{A}}(\tilde{x}_s) + G_{\mathcal{D}_1}(\tilde{x}_s) - G_{\mathcal{D}_1}(x_k) \|^2
$$

$$
= \mathbb{E} \| G_{\mathcal{A}}(x_k) - G_{\mathcal{A}}(\tilde{x}_s) - (G(x_k) - G(\tilde{x}_s)) + (G(x_k) - G(\tilde{x}_s)) + G_{\mathcal{D}_1}(\tilde{x}_s) - G_{\mathcal{D}_1}(x_k) \|^2
$$

$$
\overset{\text{①}}{\leq} 2\mathbb{E} \| G_{\mathcal{A}}(x_k) - G_{\mathcal{A}}(\tilde{x}_s) - (G(x_k) - G(\tilde{x}_s)) \|^2 + 2\mathbb{E} \| G_{\mathcal{D}_1}(\tilde{x}_s) - G_{\mathcal{D}_1}(x_k) - (G(\tilde{x}_s) - G(x_k)) \|^2
$$

$$
\overset{\text{②}}{\leq} 2 \frac{\mathbb{I}(A < n)}{A} \mathbb{E}_i \| G_i(\tilde{x}_s) - G_i(x_k) \|^2 + 2\mathbb{E} \| G_{\mathcal{D}_1}(\tilde{x}_s) - G(\tilde{x}_s) \|^2 + 2\mathbb{E} \| -G_{\mathcal{D}_1}(x_k) + G(x_k) \|^2
$$

$$
\overset{\text{③}}{\leq} 2 \frac{\mathbb{I}(A < n)}{A} B_G^2 \mathbb{E} \| x_k - \tilde{x}_s \|^2 + 4 \frac{\mathbb{I}(D < n)}{D} H_1,
$$

where ① follows from $||a+b||^2 \leq 2a^2 + 2b^2$; ② is based on Lemma 1; ③ follows from the bounded function of $G$ and the upper bound of variance of $G$. Note that $\mathcal{A}$ and $x_k$ are independent; and $\mathcal{D}$ and $\tilde{x}_s$ are independent. □

**Lemma 4.** *Suppose Assumption 1 holds, for $\hat{G}_k$ defined in (3.3) and $\nabla \tilde{f}_k$ defined in (3.4) with $\mathcal{D} = [\mathcal{D}_1, \mathcal{D}_2]$ and $D = |\mathcal{D}_1| = |\mathcal{D}_2|$, we have*

$$
\mathbb{E}_{\mathcal{D}} \|\mathbb{E}_{\mathcal{A},i_k,j_k}[\nabla \tilde{f}_k] - \nabla f(x_k)\|^2 \leq 4B_G^4 L_F^2 \left( 4 \frac{\mathbb{I}(A < n)}{A} \right) \mathbb{E}\|x_k - \tilde{x}_s\|^2
$$
$$
+ 32 B_G^2 L_F^2 \frac{\mathbb{I}(D < n)}{D} H_1 + 4 \frac{\mathbb{I}(D^2 < n^2)}{D^2} H_2.
$$

*Proof.* Through adding and subtracting the terms of

$$
(\partial G(x_k))^\mathsf{T} \nabla F(G(x_k)), (\partial G_{\mathcal{D}_1}(\tilde{x}_s))^\mathsf{T} \nabla F_{\mathcal{D}_1}(G(\tilde{x}_s)), (\partial G(\tilde{x}_s))^\mathsf{T} \nabla F(G(\tilde{x}_s)),
$$

we have

$$
\mathbb{E}_{\mathcal{D}} \|\mathbb{E}_{\mathcal{A},i_k,j_k} \left[ \nabla \tilde{f}_k \right] - \nabla f(x_k)\|^2
$$
$$
= \mathbb{E} \left\| (\partial G(x_k))^\mathsf{T} \nabla F(\hat{G}_k) - (\partial G(\tilde{x}_s))^\mathsf{T} \nabla F(G_{\mathcal{D}_1}(\tilde{x}_s)) + \nabla \hat{f}_{\mathcal{D}}(\tilde{x}_s) - \nabla f(x_k) \right\|^2
$$
$$
\overset{①}{\leq} 4 \mathbb{E} \left\| (\partial G(x_k))^\mathsf{T} \nabla F(\hat{G}_k) - (\partial G(x_k))^\mathsf{T} \nabla F(G(x_k)) \right\|^2
$$
$$
+ 4 \mathbb{E} \left\| (\partial G(\tilde{x}_s))^\mathsf{T} \nabla F(G(\tilde{x}_s)) - (\partial G(\tilde{x}_s))^\mathsf{T} \nabla F(G_{\mathcal{D}_1}(\tilde{x}_s)) \right\|^2
$$
$$
+ 4 \mathbb{E} \left\| \nabla \hat{f}_{\mathcal{D}}(\tilde{x}_s) - (\partial G_{\mathcal{D}_1}(\tilde{x}_s))^\mathsf{T} \nabla F_{\mathcal{D}_2}(G(\tilde{x}_s)) \right\|^2
$$
$$
+ 4 \mathbb{E} \left\| (\partial G_{\mathcal{D}_1}(\tilde{x}_s))^\mathsf{T} \nabla F_{\mathcal{D}_2}(G(\tilde{x}_s)) - (\partial G(\tilde{x}_s))^\mathsf{T} \nabla F(G(\tilde{x}_s)) \right\|^2
$$
$$
\overset{②}{\leq} 4B_G^2 L_F^2 \mathbb{E} \left\| \hat{G}_k - G(x_k) \right\|^2 + 4B_G^2 L_F^2 \mathbb{E}\|G(\tilde{x}_s) - G_{\mathcal{D}_1}(\tilde{x}_s)\|^2 + 4B_G^2 L_F^2 \mathbb{E}\|G(\tilde{x}_s) - G_{\mathcal{D}_1}(\tilde{x}_s)\|^2 + 4 \frac{\mathbb{I}(D^2 < n^2)}{D^2} H_2
$$
$$
\overset{③}{\leq} 4B_G^4 L_F^2 \left( 4 \frac{\mathbb{I}(A < n)}{A} \right) \mathbb{E}\|x_k - \tilde{x}_s\|^2 + 32 B_G^2 L_F^2 \frac{\mathbb{I}(D < n)}{D} H_1 + 4 \frac{\mathbb{I}(D^2 < n^2)}{D^2} H_2,
$$

where ① follows from $||a_1 + a_2 + a_3 + a_4||^2 \leq 4a_1^2 + 4a_2^2 + 4a_3^2 + 4a_4^2$; ② is based on the bounded Jacobian of $G$ and the smoothness of $F$ in Assumption 1, and the upper bound of variance in Lemma 2. ③ is based on Lemma 3 and the upper bound of variance of $G(x)$. Note that $\mathcal{A}$ and $x_k$ are independent; and $\mathcal{D}$ and $\tilde{x}_s$ are independent. □

**Lemma 5.** *Suppose Assumption 1 holds, for $\hat{G}_k$ defined in (3.3) and $\nabla \tilde{f}_k$ defined in (3.4) with $\mathcal{D} = [\mathcal{D}_1, \mathcal{D}_2]$ and $D = |\mathcal{D}_1| = |\mathcal{D}_2|$, we have*

$$
\mathbb{E}_{i_k,j_k,\mathcal{A},\mathcal{D}}\|\nabla \tilde{f}_k - \nabla f(x_k)\|^2 \leq 40 B_G^2 L_F^2 \frac{\mathbb{I}(D < n)}{D} H_1 + 5 \frac{\mathbb{I}(D^2 < n^2)}{D^2} H_2
$$
$$
+ 5 B_G^4 L_F^2 \left( \frac{L_f^2}{B_G^4 L_F^2} + 4 \frac{\mathbb{I}(A < n)}{A} \right) \mathbb{E}\|x_k - \tilde{x}_s\|^2.
$$

*Proof.* Through adding and subtracting the term of $(\partial G_j(x_k))^\mathsf{T} \nabla F_i(G(x_k))$, $(\partial G_j(\tilde{x}_s))^\mathsf{T} \nabla F_i(G(\tilde{x}_s))$, $(\partial G(\tilde{x}_s))^\mathsf{T} \nabla F(G(\tilde{x}_s))$, $(\partial G_{\mathcal{D}_1}(\tilde{x}_s))^\mathsf{T} \nabla F_{\mathcal{D}_1}(G(\tilde{x}_s))$ (Note that, $\mathcal{D}$ and $\tilde{x}_s$ are independent), we have

$$
\mathbb{E}\|\nabla \tilde{f}_k - \nabla f(x_k)\|^2
$$
$$
= \mathbb{E} \left\| (\partial G_j(x_k))^\mathsf{T} \nabla F_i(\hat{G}_k) - (\partial G_j(\tilde{x}_s))^\mathsf{T} \nabla F_i(G_{\mathcal{D}_1}(\tilde{x}_s)) + \nabla \hat{f}_{\mathcal{D}}(\tilde{x}_s) - \nabla f(x_k) \right\|^2
$$
$$
\overset{①}{\leq} 5 \mathbb{E} \left\| (\partial G_j(x_k))^\mathsf{T} \nabla F_i(G(x_k)) - (\partial G_j(\tilde{x}_s))^\mathsf{T} \nabla F_i(G(\tilde{x}_s)) - \left( \nabla f(x_k) - (\partial G(\tilde{x}_s))^\mathsf{T} \nabla F(G(\tilde{x}_s)) \right) \right\|^2
$$

$$+ 5\mathbb{E}\left\|(\partial G_j(x_k))^\mathsf{T}\nabla F_i(\hat{G}_k) - (\partial G_j(x_k))^\mathsf{T}\nabla F_i(G(x_k))\right\|^2$$

$$+ 5\mathbb{E}\left\|(\partial G_j(\tilde{x}_s))^\mathsf{T}\nabla F_i(G(\tilde{x}_s)) - (\partial G_j(\tilde{x}_s))^\mathsf{T}\nabla F_i(G_{\mathcal{D}_1}(\tilde{x}_s))\right\|^2$$

$$+ 5\mathbb{E}\left\|\nabla \hat{f}_{\mathcal{D}}(\tilde{x}_s) - (\partial G_{\mathcal{D}_1}(\tilde{x}_s))^\mathsf{T}\nabla F_{\mathcal{D}_2}(G(\tilde{x}_s))\right\|^2$$

$$+ 5\mathbb{E}\left\|(\partial G_{\mathcal{D}_1}(\tilde{x}_s))^\mathsf{T}\nabla F_{\mathcal{D}_2}(G(\tilde{x}_s)) - (\partial G(\tilde{x}_s))^\mathsf{T}\nabla F(G(\tilde{x}_s))\right\|^2$$

$$\overset{②}{\leq} 5L_f^2\mathbb{E}\|x_k - \tilde{x}_s\|^2 + 5B_G^2L_F^2\mathbb{E}\left\|\hat{G}_k - G(x_k)\right\|^2 + 5B_G^2L_F^2\mathbb{E}\|G(\tilde{x}_s) - G_{\mathcal{D}_1}(\tilde{x}_s)\|^2$$

$$+ 5B_G^2L_F^2\mathbb{E}\|G(\tilde{x}_s) - G_{\mathcal{D}_1}(\tilde{x}_s)\|^2 + 5\frac{\mathbb{I}(D^2 < n^2)}{D^2}H_2$$

$$\overset{③}{\leq} 5B_G^4L_F^2\left(\frac{L_f^2}{B_G^4L_F^2} + 4\frac{\mathbb{I}(A < n)}{A}\right)\mathbb{E}\|x_k - \tilde{x}_s\|^2 + 40B_G^2L_F^2\frac{\mathbb{I}(D < n)}{D}H_1 + 5\frac{\mathbb{I}(D^2 < n^2)}{D^2}H_2,$$

where ① follows from $\|a_1 + a_2 + a_3 + a_4 + a_5\|^2 \leq 5a_1^2 + 5a_2^2 + 5a_3^2 + 5a_4^2 + 5a_5^2$; ② is based on $\mathbb{E}[\|X - \mathbb{E}[X]\|^2] = \mathbb{E}[X^2 - \|\mathbb{E}[X]\|^2] \leq \mathbb{E}[X^2]$, the smoothness of $F_i$, the bounded Jacobian of $G(x)$ and the smoothness of $F$ in Assumption 1, and the upper bound of the variance. ③ is based on Lemma 3. $\qquad\square$

As can be seen from the above results directly, when $A$ and $D$ increase, the upper bounds are close to the bounds in (Lian et al., 2017; Liu et al., 2017a;b). Though there are extra terms with respect to $A$ and $D$, they give us another direction for analyzing the convergence rate and query complexity. The convergence rate not only depends on the convergence sequence, but also the terms including the event function $\mathbb{I}$. Thus, we can obtain the lower bound range of $A$ and $D$ that is related to $\epsilon$. Furthermore, this lemma can be applied to analyze the convergence rate and query complexity of the convex and non-convex composition problem.

## C  PROOF OF SCCG METHOD FOR COMPOSITION PROBLEM

### C.1  PROOF OF SCCG METHOD FOR STRONGLY CONVEX COMPOSITION PROBLEM

In this section, we analyze the proposed algorithm for the strongly convex composition problem. We first present the convergence of the proposed algorithm and then give the query complexity. Though the proof is similar to that of (Lian et al., 2017) and (Xiao & Zhang, 2014), we present a more clear and simple process as there is an extra term derived from the subset $\mathcal{D}$. In order to ensure the convergence of the proposed algorithm, we obtain the desired parameters' setting, such as $A$, $D$, $K$, $\eta$ and $h$. Based on the setting, we can obtain the corresponding query complexity, which is better than or equal to the SVRG-based method in (Lian et al., 2017) and (Liu et al., 2017a). This is in fact that the event function $\mathbb{I}$ has an influence on the size of $A$ and $D$.

#### C.1.1  CONVERGENCE ANALYSIS

Based on the strong convex and smoothness of the function of $f(x)$, we provide the convergence sequence, in which the parameters are not defined. But the sequences motivate us to consider the parameters' setting such that lead to the desired convergence rate. Note that, $\mathcal{D}$ and $\tilde{x}_s$ are independent.

**Theorem 3.** *Suppose Assumption 1 holds, in Algorithm 1, let $h > 0, \eta > 0$, $A = |\mathcal{A}|$, $D = |\mathcal{D}_1| = |\mathcal{D}_2|$, $K$ is the number of the inner iteration, $x^*$ is the optimal point, we have*

$$\mathbb{E}\|\tilde{x}_S - x^*\|^2 \leq \rho^S\mathbb{E}\|\tilde{x}_0 - x^*\|^2 + \frac{\rho_3}{\rho_1}\frac{1 - \rho^S}{1 - \rho},$$

*where $\rho = (\frac{1}{K} + \rho_2)/\rho_1$, $\rho_2$ and $\rho_3$ defined ($V$, $V_1$ are defined in (C.4) and (C.5).)*

$$\rho_1 = \left(2\mu - h - 4V\frac{1}{h} - (12L_f^2 + 10V)\eta\right)\eta, \qquad (C.1)$$

$$\rho_2 = 2\left(2V\frac{1}{h} + 5\left(L_f^2 + V\right)\eta\right)\eta, \tag{C.2}$$

$$\rho_3 = \frac{1}{h}\eta\frac{4}{5}V_1 + 2\eta^2 V_1. \tag{C.3}$$

*Proof.* By the update of $x_k$ in Algorithm 1, we have

$$\mathbb{E}\|x_{k+1} - x^*\|^2$$

$$= \mathbb{E}\|x_k - x^*\|^2 - 2\eta\mathbb{E}\langle\nabla\tilde{f}_k, x_k - x^*\rangle + \eta^2\mathbb{E}\left\|\nabla\hat{f}_k\right\|^2$$

$$= \mathbb{E}\|x_k - x^*\|^2 - 2\eta\mathbb{E}\langle\nabla f(x_k) + \mathbb{E}_{\mathcal{A},i,j}\left[\nabla\tilde{f}_k\right] - \nabla f(x_k), x_k - x^*\rangle + \eta^2\mathbb{E}\left\|\nabla\hat{f}_k\right\|^2$$

$$= \mathbb{E}\|x_k - x^*\|^2 - 2\eta\mathbb{E}\langle\nabla f(x_k), x_k - x^*\rangle - 2\eta\mathbb{E}\langle\mathbb{E}_{\mathcal{A},i,j}\left[\nabla\tilde{f}_k\right] - \nabla f(x_k), x_k - x^*\rangle$$

$$\quad + \eta^2\mathbb{E}\left\|\nabla\hat{f}_k + \nabla f(x_k) - \nabla f(x_k)\right\|^2$$

$$\overset{\text{①}}{\leq}\mathbb{E}\|x_k - x^*\|^2 - 2\eta\mu\mathbb{E}\|x_k - x^*\|^2 + \eta\frac{1}{h}\mathbb{E}\left\|\mathbb{E}_{\mathcal{A},i,j}\left[\nabla\tilde{f}_k\right] - \nabla f(x_k)\right\|^2 + h\eta\mathbb{E}\|x_k - x^*\|^2$$

$$\quad + 2\eta^2\left(\mathbb{E}\|\nabla f(x_k)\|^2 + \mathbb{E}\left\|\nabla\tilde{f}_k - \nabla f(x_k)\right\|^2\right)$$

$$= \mathbb{E}\|x_k - x^*\|^2 - (2\eta\mu - h\eta)\mathbb{E}\|x_k - x^*\|^2 + \eta\frac{1}{h}\mathbb{E}\left\|\mathbb{E}_{\mathcal{A},i,j}\left[\nabla\tilde{f}_k\right] - \nabla f(x_k)\right\|^2$$

$$\quad + 2\eta^2\left(\mathbb{E}\|\nabla f(x_k) - \nabla f(x^*)\|^2 + \mathbb{E}\left\|\nabla\tilde{f}_k - \nabla f(x_k)\right\|^2\right)$$

$$\overset{\text{②}}{\leq}\mathbb{E}\|x_k - x^*\|^2 - (2\eta\mu - h\eta)\mathbb{E}\|x_k - x^*\|^2 + \eta\frac{1}{h}\left(4V\|x_k - \tilde{x}_s\|^2 + V_2\right)$$

$$\quad + 2\eta^2\left(L_f^2\mathbb{E}\|x_k - x^*\|^2 + 5\left(L_f^2 + V\right)\|x_k - \tilde{x}_s\|^2 + V_1\right)$$

$$= \mathbb{E}\|x_k - x^*\|^2 - \left(2\mu - h - 4V\frac{1}{h} - \left(12L_f^2 + 10V\right)\eta\right)\eta\mathbb{E}\|x_k - x^*\|^2$$

$$\quad + 2\left(2V\frac{1}{h} + 5\left(L_f^2 + V\right)\eta\right)\eta\mathbb{E}\|\tilde{x}_s - x^*\|^2 + \frac{1}{h}\eta V_2 + 2\eta^2 V_1,$$

where

$$V = B_G^4 L_F^2\left(4\frac{\mathbb{I}(A < n)}{A}\right), \tag{C.4}$$

$$V_1 = 40B_G^2 L_F^2\frac{\mathbb{I}(D < n)}{D}H_1 + 5\frac{\mathbb{I}(D^2 < n^2)}{D^2}H_2, \tag{C.5}$$

$$V_2 = 32B_G^2 L_F^2\frac{\mathbb{I}(D < n)}{D}H_1 + 4\frac{\mathbb{I}(D^2 < n^2)}{D^2}H_2 = \frac{4}{5}V_1, \tag{C.6}$$

① is based on $\|a_1 + a_2\|^2 \leq 2a_1^2 + 2a_2^2$ and $\langle a_1, a_2\rangle \leq h\|a_1\|^2 + \frac{1}{h}\|a_2\|^2, h > 0$; ② is based on strongly-convex of $f$ in Assumption 1, and Lemma 4, 5.

Summing up from $k = 0$ to $k = K - 1$, we have

$$E\|x_K - x^*\|^2 \leq E\|x_0 - x^*\|^2 - \rho_1\sum_{k=0}^{K-1}E\|x_k - x^*\|^2 + K\rho_2 E\|\tilde{x}_s - x^*\| + K\rho_3,$$

where

$$\rho_1 = \left(2\mu - h - 4V\frac{1}{h} - \left(12L_f^2 + 10V\right)\eta\right)\eta,$$

$$\rho_2 = 2\left(2V\frac{1}{h} + 5\left(L_f^2 + V\right)\eta\right)\eta,$$

$$\rho_3 = \frac{1}{h}\eta V_2 + 2\eta^2 V_1.$$

For $x_0 = \tilde{x}_s$, by arranging, we have

$$\rho_1 \mathbb{E}\|\tilde{x}_{s+1} - x^*\|^2 \leq \frac{1}{K}\mathbb{E}\|x_0 - x^*\|^2 + \rho_2 \mathbb{E}\|\tilde{x}_s - x^*\|^2 + \rho_3 - \frac{1}{K}\mathbb{E}\|x_K - x^*\|^2$$

$$\leq \left(\frac{1}{K} + \rho_2\right)\mathbb{E}\|\tilde{x}_s - x^*\|^2 + \rho_3.$$

we assume that $\rho_1 > 0$ in (C.1), then we can obtain

$$\mathbb{E}\|\tilde{x}_S - x^*\|^2 \leq \rho^S \mathbb{E}\|\tilde{x}_0 - x^*\|^2 + \frac{\rho_3}{\rho_1}\sum_{s=0}^{S}\rho^s$$

$$\leq \rho^S \mathbb{E}\|\tilde{x}_0 - x^*\|^2 + \frac{\rho_3}{\rho_1}\frac{1-\rho^S}{1-\rho}, \qquad \text{(C.7)}$$

where $\rho = (\frac{1}{K} + \rho_2)/\rho_1$, $\rho_2$ and $\rho_3$ defined in (C.2) and (C.3), the last inequality is based on the formula of geometric progression. $\qquad \square$

Thus, if $\tilde{x}_S$ converges to the optimal point $x^*$, we need to ensure that $\rho < 1$ and the second term $\rho_3(1-\rho^S)/(\rho_1(1-\rho))$ is less than $\epsilon/2$. Actually, if $D = n$, the second term is equal to zero, which will be similar to the convergence results in (Lian et al., 2017) and (Liu et al., 2017a).

**Proof of Theorem 1**

*Proof.* In order to keep the proposed algorithm converge, we consider the parameters' setting, we first ensure that $\rho_1 > 0$ in (C.1), and then define

$$\rho = (\frac{1}{K} + \rho_2)/\rho_1, \qquad \text{(C.8)}$$

that require $\rho < 1$, where $\rho_2$ defined in (C.2). Thus, the convergence sequence is

$$\mathbb{E}\|\tilde{x}_S - x^*\|^2 \leq \rho^S \mathbb{E}\|\tilde{x}_0 - x^*\|^2 + \frac{\rho_3}{\rho_1}\sum_{s=0}^{S}\rho^s \leq \rho^S \mathbb{E}\|\tilde{x}_0 - x^*\|^2 + \frac{\rho_3}{\rho_1}\frac{1}{1-\rho}.$$

We ensure $\frac{\rho_3}{\rho_1}\frac{1}{1-\rho} \leq \frac{1}{2}\epsilon$, where $\rho_3$ defined in (C.3), that we can derive the size of the $D$. In the following we analyze the parameters' setting such that satisfying the above requirement.

1. In order to ensure $\rho_1 > 0$ in (C.1), we consider the parameter $h$, $\eta$ and $A$,

   (a) $h = \mu$, consider $\rho_1$ in (C.1), we should require that $h \leq \mu$, however, V in (C.4) has the relationship with $A$ and $D$. In order to keep $A$ small enough, we set the upper bound of $h$. Thus, we set $h = \mu$.

   (b) $A = \min\left\{n, 128B_G^4 L_F^2 \frac{1}{\mu^2}\right\}$, based on the setting of h, we require that $V/h < \frac{\mu}{16}$. Thus, we have

   $$V = B_G^4 L_F^2\left(4\frac{I(A<n)}{A}\right) \leq 8B_G^4 L_F^2 \frac{I(A<n)}{A} \leq \frac{1}{16}\mu^2.$$

   For V defined in (C.4), if $A < n$, we have

   $$A \geq 128B_G^4 L_F^2 \frac{1}{\mu^2},$$

   otherwise, $A = n$ satisfy the requirement. Thus, we have $A = \min\left\{n, 128B_G^4 L_F^2 \frac{1}{\mu^2}\right\}$.

   (c) $\eta \leq \frac{3\mu}{53L_f^2}$, back to the target of $\rho_1 > 0$, we require that $\eta \leq \frac{3\mu}{53L_f^2} \leq \frac{\frac{3}{4}\mu}{12L_f^2 + \frac{10}{8}L_f^2} \leq \frac{\frac{3}{4}\mu}{12L_f^2 + \frac{10}{8}\mu^2} = \frac{\mu - 4\frac{1}{\mu}V}{12L_f^2 + 10V} = \frac{2\mu - h - 4\frac{1}{h}V}{2L_f^2 + 10(L_f^2 + V)}$, note that $\mu \leq L_f$ by the definition in preliminaries.

2. In order to ensure $\rho < 1$ in (C.8), we first consider $\rho_1$ and $\rho_2$ in (C.1) and (C.2). By the setting of $h = \mu$ and $V < \mu^2/16$, we have,

$$\rho_1 \geq \left( \mu - 2L_f^2 \eta - \left( \frac{1}{4}\mu + 10 \left( L_f^2 + \frac{1}{16}\mu^2 \right) \eta \right) \right) \eta \geq \left( \frac{3}{4}\mu - \frac{101}{8} L_f^2 \eta \right) \eta, \qquad \text{(C.9)}$$

$$\rho_2 \leq 4\frac{1}{\mu}\eta\frac{1}{16}\mu^2 + 10 \left( L_f^2 + \frac{1}{16}\mu^2 \right) \eta^2 \leq \left( \frac{1}{4}\mu + 10 \left( L_f^2 + \frac{1}{16}\mu^2 \right) \eta \right) \eta \geq \left( \frac{1}{4}\mu + \frac{85}{8} L_f^2 \eta \right) \eta. \qquad \text{(C.10)}$$

We require that $\rho = \frac{1}{K\rho_1} + \frac{\rho_2}{\rho_1} < 1$, and analyze the two terms separately,

(a) In order to $\frac{\rho_2}{\rho_1} < \frac{1}{2}$, that is

$$\frac{\rho_2}{\rho_1} < \frac{\left( \frac{1}{4}\mu + \frac{85}{8} L_f^2 \eta \right) \eta}{\left( \frac{3}{4}\mu - \frac{101}{8} L_f^2 \eta \right) \eta} < \frac{1}{2}.$$

We get $\eta \leq \frac{\mu}{135 L_f^2}$.

(b) In order to $\frac{1}{K\rho_1} < \frac{1}{2}$, that is

$$\frac{1}{K\rho_1} < \frac{1}{2K\rho_2} \leq \frac{1}{2K \left( \frac{1}{4}\mu + 10 \left( L_f^2 + \frac{1}{16}\mu^2 \right) \eta \right) \eta}$$

$$\leq \frac{1}{2K \left( \frac{1}{4}\mu + \frac{85}{8} L_f^2 \eta \right) \eta} \leq \frac{1}{2K \left( \frac{1}{4}\mu\eta \right)} < \frac{1}{2}.$$

Thus, we have $K \geq 540 \frac{L_f^2}{\mu^2}$.

3. Consider the term $\rho^S E \|\tilde{x}_0 - x^*\|^2 + \frac{\rho_3}{\rho_1} \frac{1}{1-\rho}$, we analyze them separately,

(a) In order to ensure $\frac{\rho_3}{\rho_1} \frac{1}{1-\rho} \leq \frac{1}{2}\epsilon$, that is

$$\frac{\rho_3}{\rho_1} \frac{1}{1 - \left( \frac{1}{K\rho_1} + \frac{\rho_2}{\rho_1} \right)} = \frac{\rho_3}{\rho_1 - \frac{1}{K} - \rho_2} \leq \frac{\rho_3}{\rho_1 - \frac{1}{K} - \frac{1}{2}\rho_1} \leq \frac{\rho_3}{\frac{1}{2}\rho_1 - \frac{1}{K}} \leq \frac{2\rho_3}{\rho_1} \leq \frac{1}{2}\epsilon.$$

Based on the bound of $\rho_1$ in (C.9), the definition of $V_1$ in (C.5) and the step size $\eta$ mentioned above, we have

i. For $V$

$$2\frac{\frac{1}{\mu}\eta V_2 + 2\eta^2 V_1}{\rho_1} = 2\frac{\frac{1}{\mu}V_2 + 2\eta V_1}{\frac{3}{4}\mu - \frac{101}{8} L_f^2 \eta} = \frac{\frac{4}{5}\frac{1}{\mu}V_1 + 2\eta V_1}{\frac{3}{4}\mu - \frac{101}{8} L_f^2 \eta} = \frac{\left( \frac{4}{5\mu} + 2\eta \right) V_1}{\frac{3}{4}\mu - \frac{101}{8} L_f^2 \eta} \leq \epsilon,$$

thus, we have

$$V_1 \leq \frac{4}{5}\epsilon\mu^2 \leq \frac{\left( \frac{3}{4} - \frac{101}{8}\frac{1}{135} \right)}{\frac{4}{5} + \frac{2}{135}} \epsilon\mu^2 \leq \frac{\left( \frac{3}{4} - \frac{101}{8}\frac{1}{135} \right)\mu}{\frac{4}{5\mu} + 2\frac{\mu}{135\mu^2}}\epsilon \leq \frac{\frac{3}{4}\mu - \frac{101}{8}L_f^2\frac{\mu}{135L_f^2}}{\frac{4}{5\mu} + 2\frac{\mu}{135L_f^2}}\epsilon \leq \frac{\frac{3}{4}\mu - \frac{101}{8}L_f^2\eta}{\left( \frac{4}{5\mu} + 2\eta \right)}\epsilon$$

ii. If $D < n$, we can obtain $D \geq \frac{5}{4\epsilon\mu^2} \left( 20B_G^4 L_F^2 H_1 + 5H_2 \right)$, otherwise $D = 0$, the above inequality is correct. Thus, we obtain $D = \min \left\{ n, \left( 16B_G^4 L_F^2 H_1 + 4H_2 \right) \frac{5}{4\epsilon\mu^2} \right\}$.

(b) In order to ensure $\rho^S E \|\tilde{x}_0 - x^*\|^2 \leq \frac{1}{2}\epsilon$, we need the number of the outer iterations

$$S \geq \frac{1}{\log (1/\rho)} \log \frac{2E\|\tilde{x}_0 - x^*\|^2}{\epsilon}.$$

All in all, we consider the query complexity based on above parameters' setting. For each outer iteration, there will be $(D + KA)$ queries. Thus, the query complexity is

$$(D + KA) S = \mathcal{O}\left(\left(\min\left\{n, \frac{1}{\epsilon\mu^2}\right\} + \frac{L_f^2}{\mu^2}\min\left\{n, \frac{1}{\mu^2}\right\}\right)\log(1/\epsilon)\right).$$

$\square$

## C.2 PROOF OF SCCG METHOD FOR NON-CONVEX COMPOSITION PROBLEM

### C.2.1 RELATED BOUNDS

**Lemma 6.** *Suppose Assumption 1 hold, in Algorithm 1, we can obtain the following new sequence with respect to $f(x_k)$ and $\|x_k - \tilde{x}_s\|^2$, let $h > 0, \eta > 0$, $A = |\mathcal{A}|$ and $D = |\mathcal{D}_1| = |\mathcal{D}_2|$, we have*

$$\mathbb{E}[f(x_{k+1})] + c_{k+1}\mathbb{E}\|x_{k+1} - \tilde{x}_s\|^2 \leq \mathbb{E}[f(x_k)] + c_k\mathbb{E}\|x_k - \tilde{x}_s\|^2 - u_k\|\nabla f(x_k)\|^2 + J_k,$$

*where*

$$W = B_G^4 L_F^2 \left(4\frac{\mathbb{I}(A < n)}{A} + 4\frac{\mathbb{I}(D < n)}{D}\right), \tag{C.11}$$

$$c_k = c_{k+1}\left(1 + \left(\frac{2}{h} + 4hW\right)\eta + 10\left(L_f^2 + W\right)\eta^2\right)$$
$$+ 2W\eta + 5(L_f^2 + W)L_f\eta^2, \tag{C.12}$$

$$u_k = \left(\left(\frac{1}{2} - hc_{k+1}\right)\eta - (L_f + 2c_{k+1})\eta^2\right), \tag{C.13}$$

$$W_1 = 20B_G^2 L_F^2 \frac{\mathbb{I}(D < n)}{D}H_1 + 5\frac{\mathbb{I}(D^2 < n^2)}{D^2}H_2, \tag{C.14}$$

$$J_k = \left(\frac{1}{2} + hc_{k+1}\right)\frac{4}{5}W_1\eta + (L_f + 2c_{k+1})W_1\eta^2. \tag{C.15}$$

*Proof.* Consider the upper bound of $f(x_{k+1})$ and $\|x_{k+1} - \tilde{x}_s\|^2$, respectively,

- Base on the smoothness of $f$ in Assumption 1 and take expectation with respective to $i_k, j_k$, we have

$$\mathbb{E}_{i,j}\left[f(x_{k+1})\right]$$
$$\leq \mathbb{E}\left[f(x_k)\right] - \eta\mathbb{E}\langle\nabla f(x_k), \nabla\tilde{f}_k\rangle + \frac{L_f}{2}\eta^2\mathbb{E}\left\|\nabla\tilde{f}_k\right\|^2$$
$$= \mathbb{E}\left[f(x_k)\right] - \eta\mathbb{E}\langle\nabla f(x_k), \nabla\tilde{f}_k - \nabla f(x_k) + \nabla f(x_k)\rangle + \frac{L_f}{2}\eta^2\mathbb{E}\left\|\nabla\tilde{f}_k\right\|^2$$
$$= \mathbb{E}\left[f(x_k)\right] - \eta\mathbb{E}\langle\nabla f(x_k), \nabla f(x_k)\rangle - \eta\langle\nabla f(x_k), \mathbb{E}\left[\nabla\tilde{f}_k\right] - \nabla f(x_k)\rangle + \frac{L_f}{2}\eta^2\mathbb{E}\left\|\nabla\tilde{f}_k - \nabla f(x_k) + \nabla f(x_k)\right\|^2$$
$$\leq \mathbb{E}\left[f(x_k)\right] - \eta\mathbb{E}\|\nabla f(x_k)\|^2 + \frac{1}{2}\eta\mathbb{E}\|\nabla f(x_k)\|^2 + \frac{1}{2}\eta\mathbb{E}\left\|\mathbb{E}_{\mathcal{A},i,j}\left[\nabla\tilde{f}_k\right] - \nabla f(x_k)\right\|^2$$
$$+ \frac{L_f}{2}\eta^2\left(2\mathbb{E}\|\nabla f(x_k)\|^2 + 2\mathbb{E}\left\|\nabla\tilde{f}_k - \nabla f(x_k)\right\|^2\right)$$
$$= \mathbb{E}\left[f(x_k)\right] - \frac{1}{2}\eta\mathbb{E}\|\nabla f(x_k)\|^2 + \frac{1}{2}\eta\mathbb{E}\left\|\mathbb{E}_{\mathcal{A},i,j}\left[\nabla\tilde{f}_k\right] - \nabla f(x_k)\right\|^2 + L_f\eta^2\left(\mathbb{E}\|\nabla f(x_k)\|^2 + \mathbb{E}\left\|\nabla\tilde{f}_k - \nabla f(x_k)\right\|^2\right)$$
$$= \mathbb{E}\left[f(x_k)\right] - \left(\frac{1}{2}\eta - L_f\eta^2\right)\mathbb{E}\|\nabla f(x_k)\|^2 + \frac{1}{2}\eta\mathbb{E}\left\|\mathbb{E}_{\mathcal{A},i,j}\left[\nabla\tilde{f}_k\right] - \nabla f(x_k)\right\|^2 + L_f\eta^2\mathbb{E}\left\|\nabla\tilde{f}_k - \nabla f(x_k)\right\|^2,$$

where the last inequality is based on $\|a_1 + a_2\|^2 \leq 2a_1^2 + 2a_2^2$.

- Base on the update of $x_k$ in Algorithm 1 and take expectation with respective to $i_k, j_k$, we have,

$$\mathbb{E}_{i,j}\|x_{k+1} - \tilde{x}_s\|^2$$

$$=\mathbb{E}\|x_k - \tilde{x}_s\|^2 - 2\eta\mathbb{E}\langle\nabla\tilde{f}_k, x_k - \tilde{x}_s\rangle + \eta^2\mathbb{E}\left\|\nabla\tilde{f}_k\right\|^2$$

$$=\mathbb{E}\|x_k - \tilde{x}_s\|^2 - 2\eta\mathbb{E}\langle\nabla\tilde{f}_k - \nabla f(x_k) + \nabla f(x_k), x_k - \tilde{x}_s\rangle] + \eta^2\mathbb{E}\left\|\nabla\tilde{f}_k\right\|^2$$

$$=\mathbb{E}\|x_k - \tilde{x}_s\|^2 - 2\eta\mathbb{E}\langle\nabla f(x_k), x_k - \tilde{x}_s\rangle] - 2\eta\langle\mathbb{E}\left[\nabla\tilde{f}_k\right] - \nabla f(x_k), x_k - \tilde{x}_s\rangle]$$

$$\quad + \eta^2\mathbb{E}\left\|\nabla\tilde{f}_k - \nabla f(x_k) + \nabla f(x_k)\right\|^2$$

$$\leq\mathbb{E}\|x_k - \tilde{x}_s\|^2 + h\eta\|\nabla f(x_k)\|^2 + h\eta\left\|\mathbb{E}\left[\nabla\tilde{f}_k\right] - \nabla f(x_k)\right\|^2 + \frac{2}{h}\eta\mathbb{E}\|x_k - \tilde{x}_s\|^2$$

$$\quad + \eta^2\left(2\mathbb{E}\|\nabla f(x_k)\|^2 + 2\mathbb{E}\left\|\nabla\tilde{f}_k - \nabla f(x_k)\right\|^2\right)$$

$$=\left(1 + \frac{2}{h}\eta\right)\mathbb{E}\|x_k - \tilde{x}_s\|^2 + \left(h\eta + 2\eta^2\right)\mathbb{E}\|\nabla f(x_k)\|^2 + h\eta\mathbb{E}\left\|\mathbb{E}\left[\nabla\tilde{f}_k\right] - \nabla f(x_k)\right\|^2 + 2\eta^2\mathbb{E}\left\|\nabla\tilde{f}_k - \nabla f(x_k)\right\|^2,$$

where the inequality is based on $2\langle a_1, b_2\rangle \leq 1/h\|a_1\|^2 + h\|a_2\|^2, \forall h > 0$, and $\|a_1 + a_2\|^2 \leq 2a_1^2 + 2a_2^2$.

Combine above equalities and Lemma 4, 5, we form a Lyapunov function,

$$\mathbb{E}[f(x_{k+1})] + c_{k+1}\mathbb{E}\|x_{k+1} - \tilde{x}_s\|^2$$

$$=\mathbb{E}[f(x_k)] - \left(\frac{1}{2}\eta - L_f\eta^2\right)\|\nabla f(x_k)\|^2 + \frac{1}{2}\eta\left\|\mathbb{E}\left[\nabla\tilde{f}_k\right] - \nabla f(x_k)\right\|^2 + L_f\eta^2\left\|\nabla\tilde{f}_k - \nabla f(x_k)\right\|^2$$

$$\quad + c_{k+1}\left(\left(1 + \frac{2}{h}\eta\right)\mathbb{E}\|x_k - \tilde{x}_s\|^2 + \left(h\eta + 2\eta^2\right)\|\nabla f(x_k)\|^2 + h\eta\left\|\mathbb{E}\left[\nabla\tilde{f}_k\right] - \nabla f(x_k)\right\|^2 + 2\eta^2\left\|\nabla\tilde{f}_k - \nabla f(x_k)\right\|^2\right)$$

$$=\mathbb{E}[f(x_k)] + c_{k+1}\left(1 + \frac{2}{h}\eta\right)\mathbb{E}\|x_k - \tilde{x}_s\|^2 - \left(\left(\frac{1}{2} - c_{k+1}h\right)\eta - \left(L_f + 2c_{k+1}\right)\eta^2\right)\|\nabla f(x_k)\|^2$$

$$\quad + \left(L_f\eta^2 + 2\eta^2 c_{k+1}\right)\left\|\nabla\tilde{f}_k - \nabla f(x_k)\right\|^2 + \left(\frac{1}{2}\eta + h\eta c_{k+1}\right)\left\|\mathbb{E}\left[\nabla\tilde{f}_k\right] - \nabla f(x_k)\right\|^2$$

$$\leq\mathbb{E}[f(x_k)] + c_k\mathbb{E}\|x_k - \tilde{x}_s\|^2 - u_k\|\nabla f(x_k)\|^2 + J_k,$$

where

$$u_k = \left(\left(\frac{1}{2} - hc_{k+1}\right)\eta - \left(L_f + 2c_{k+1}\right)\eta^2\right);$$

$$W_1 = 40B_G^2 L_F^2\frac{\mathbb{I}(D < n)}{D}H_1 + 5\frac{\mathbb{I}(D^2 < n^2)}{D^2}H_2;$$

$$W_2 = \frac{4}{5}W_1;$$

$$J_k = \left(\frac{1}{2} + hc_{k+1}\right)W_2\eta + \left(L_f + 2c_{k+1}\right)W_1\eta^2;$$

$$W = B_G^4 L_F^2\left(4\frac{\mathbb{I}(A < n)}{A}\right);$$

$$c_k = c_{k+1}\left(1 + \left(\frac{2}{h} + 4hW\right)\eta + 10\left(L_f^2 + W\right)\eta^2\right) + 2W\eta + 5(L_f^2 + W)L_f\eta^2.$$

$\square$

Based on the above inequality with respect to the sequence $\mathbb{E}[f(x_k)] + c_k\mathbb{E}\|x_k - \tilde{x}_s\|^2$ and Algorithm 1, we can obtain the convergence form in which the parameters are not clear defined.

**Theorem 4.** *In Algorithm 1, suppose Assumption 1 holds, we can obtain the following new sequence with respect to $f(x_k)$ and $||x_k - \tilde{x}_s||^2$. $K$ is the number of inner iterations, $S$ is the number of inner iterations, we have*

$$u_0\mathbb{E}[\|\nabla f(\hat{x}_k^s)\|^2] \leq \frac{f(x_0) - f(x^*)}{KS} + J_0,$$

*where $\hat{x}_k^s$ is the output point, $J_0$ and $u_0$ are defined in (C.15) and (C.13).*

*Proof.* Based on the update for $c_k$ in (C.12), we can see that $c_k > c_{k+1}$. As $c_k$ is a decreasing sequence, we have $u_0 < u_k$ and $J_k < J_0$. Then, we get

$$u_0\mathbb{E}[\|\nabla f(x_k)\|^2] \leq E[f(x_k)] + c_k\mathbb{E}[\|x_k - \tilde{x}_s\|^2] - (\mathbb{E}[f(x_{k+1})] + c_{k+1}\mathbb{E}[\|x_{k+1} - \tilde{x}_s\|^2]) + J_0.$$

Sum from $k = 0$ to $k = K - 1$, we can get

$$\frac{1}{K}\sum_{k=0}^{K-1} u\mathbb{E}[\|\nabla f(x_k)\|^2] \leq \frac{\mathbb{E}[f(x_0)] - (\mathbb{E}[f(x_K)] + c_K\mathbb{E}[\|x_K - \tilde{x}_s\|^2])}{K} + J_0$$

$$\leq \frac{\mathbb{E}[f(x_0)] - \mathbb{E}[f(x_K)]}{K} + J_0.$$

Since $x_0 = \tilde{x}_s$, let $\tilde{x}_{s+1} = x_K$, we obtain,

$$\frac{1}{K}\sum_{k=0}^{K-1} u_0\mathbb{E}[\|\nabla f(x_k)\|^2] \leq \frac{\mathbb{E}[f(\tilde{x}_s)] - \mathbb{E}[f(\tilde{x}_{s+1})]}{K} + J_0.$$

Summing the outer iteration from $s = 0$ to $S - 1$, we have

$$u_0\mathbb{E}[\|\nabla f(\hat{x}_k^s)\|^2] = \frac{1}{S}\sum_{s=0}^{S-1}\frac{1}{K}\sum_{k=0}^{K-1} u_0\mathbb{E}[\|\nabla f(x_k^s)\|^2] + J_0$$

$$\leq \frac{\mathbb{E}[f(\tilde{x}_0)] - \mathbb{E}[f(\tilde{x}_S)]}{KS} + J_0 \leq \frac{f(x_0) - f(x^*)}{KS} + J_0,$$

where $x_k^s$ indicates the $s$-th outer iteration at $k$-th inner iteration, and $\hat{x}_k^s$ is uniformly and randomly chosen from $s = \{0, ..., S - 1\}$ and k=$\{0, .., K - 1\}$. $\square$

### C.2.2 CONVERGENCE ANALYSIS

Base on Algorithm 1, the analysis of convergence is based on the smoothness of $f(x)$ and the update of $x$ under the Lyapunov function to form the convergence sequence. Theorem 1 shows that our proposed algorithm can converge to the stationary point.

The convergence proof is similar to that of (Liu et al., 2017b; Reddi et al., 2016), however, our algorithm considers the inexact computation of the gradient at the beginning of each epoch. Thus, we derive the different parameters' setting. In particular, the number of the subset $\mathcal{D}$ and $\mathcal{A}$ depend on the *min* function. Intuitively, we can compute the gradient and inner function based on the subset rather on the whole sample. Moreover, considering the convergence results, we can see that the step size $\eta$ has the relationship with many parameters, such as the subset $\mathcal{A}$, inner iteration $K$ and the total iteration $T$.

**Proof of Theorem 2**

*Proof.* In order to have $\mathbb{E}[\|\nabla f(\hat{x}_k^s)\|^2] \leq \epsilon$, that is

$$\mathbb{E}[\|\nabla f(\hat{x}_k^s)\|^2] \leq \frac{L_f(f(x_0) - f(x^*))}{u_0 SK} + J_0/u_0 \leq \frac{\epsilon}{2} + \frac{\epsilon}{2} \leq \epsilon,$$

we consider the corresponding parameters' setting:

1. For the first term, consider $c_k$ defined in (C.12) define $c_k = c_{k+1}Y + U$, for $k = K$, we have

$$c_K = \left(\frac{1}{Y}\right)^K \left(c_0 + \frac{U}{Y-1}\right) - \frac{U}{Y-1},$$

where

$$Y = 1 + \left(\frac{2}{h} + 4hW\right)\eta + 10\left(B_G^4 L_F^2 + W\right)\eta^2,$$

$$U = 2W\eta + 5(L_f^2 + W)L_f\eta^2 > 0.$$

By setting $c_K \to 0$, we obtain

$$c_0 = \frac{UY^K}{Y-1} - \frac{U}{Y-1} = \frac{U\left(Y^K - 1\right)}{Y-1}.$$

Then, putting the Y and U into the above equation. We have

$$c_0 = \frac{2W\eta + 5(L_f^2 + W)L_f\eta^2}{\left(\frac{2}{h} + 4hW\right)\eta + 10\left(L_f^2 + W\right)\eta^2}C = \frac{2W + 5(L_f^2 + W)L_f\eta}{\left(\frac{2}{h} + 4hW\right) + 10\left(L_f^2 + W\right)\eta}C, \text{ (C.16)}$$

where $C = Y^K - 1$. Because $c_0$ has the influence on the parameters such as $K$, $C$ and $u_0$, we analyze them separately,

(a) For $K$ and $C$, based on the character of function $\left(1 + \frac{1}{t_2}\right)^{t_1} \to e$,[6] as $t_1, t_2 \to +\infty$ and $t_1 t_2 < 1$, and the function is also the increasing function with an upper bound of $e$, we require

$$K < 1/\left(\left(\frac{2}{h} + 4hW\right)\eta + 10\left(L_f^2 + W\right)\eta^2\right), \quad \text{(C.17)}$$

thus, we have $C < e - 1$.

(b) For $u_0$ defined in (C.13), in order to keep $u_k > 0$, we need to keep $c_0 h < 1/4$. If $c_0 h < 1/4$, there exits a constant $\tilde{u}$ such that $u_0 = \tilde{u}\eta$. In order to satisfy $c_0 h < 1/4$, combine with (C.16) and $C < e - 1$, that is

$$c_0 h \le \frac{2W + 5(L_f^2 + W)L_f\eta}{\left(\frac{2}{h} + 4hW\right) + 10\left(L_f^2 + W\right)\eta}h\left(e - 1\right) \le \frac{1}{4},$$

i. By setting $h = \frac{1}{5\sqrt{L_f^3\eta}}$, there exist $\tilde{w} > 0$, based on above inequality, we have

$$W \le \frac{16L_f^3\eta + 50L_f^{3.5}\sqrt{\eta}\eta}{9.6 + 34L_f^3\eta - 50\sqrt{L_f^3\eta}\eta} < \tilde{w}L_f^3\eta$$

Thus, combine with the definition of W in (C.11), we require that

$$W = B_G^4 L_F^2\left(4\frac{\mathbb{I}(A < n)}{A}\right) \le 8B_G^4 L_F^2\frac{\mathbb{I}(A < n)}{A} \le \tilde{w}L_f^3\eta = \mathcal{O}\left(L_f^3\eta\right).$$

If $A < n$, we require $A \ge \mathcal{O}\left(B_G^4 L_F^2/(L_f^3\eta)\right)$. Thus, we have $A = \min\left\{n, \mathcal{O}\left(1/\eta\right)\right\}$.

ii. Based on the setting of $h$ and $W$, combing with (C.17), we have

$$K < \frac{1}{\left(10\sqrt{L_f^3\eta} + \frac{4}{5\sqrt{L_f^3\eta}}\tilde{w}L_f^3\eta\right)\eta + 10\left(L_f^2 + \tilde{w}L_f^3\eta\right)\eta^2}$$

$$= \frac{1}{\left(10\sqrt{L_f^3\eta} + \frac{4}{5}\sqrt{L_f^3\eta}\right)\eta + 10\left(L_f^2 + \eta\right)\eta^2} = \mathcal{O}\left(\frac{1}{(L_f\eta)^{3/2}}\right).$$

---

[6]Here the 'e' is the Euler number, approximate to 2.718.

2. For the second term about $J_0$, as $u_0 = w_1\eta$, we require

$$\frac{J_0}{\tilde{u}\eta} = \frac{1}{\tilde{u}}\left(\frac{1}{2} + hc_0\right)W_2 + (L_f + 2c_0)W_1\eta$$

$$\leq \frac{1}{\tilde{u}}W_1\left(\frac{3}{5} + L_f\eta + \frac{1}{2}\eta\sqrt{\eta}\right)$$

$$\leq \frac{1}{\tilde{u}}\left(20B_G^2L_F^2H_1 + 5H_2\right)\left(\frac{3}{5} + L_f\eta + \eta\sqrt{\eta}\right)\frac{\mathbb{I}(D < n)}{D} \leq \frac{1}{2}\epsilon,$$

Then, if $D < n$, we require that

$$D \geq \frac{2}{\epsilon\tilde{u}}\left(20B_G^2L_F^2H_1 + 5H_2\right)\left(\frac{3}{5} + \frac{1}{2}L_f\eta + c_0\eta\sqrt{\eta}\right) = \mathcal{O}\left(\frac{1}{\epsilon}\right).$$

Thus, we set $D = \min\{n, \mathcal{O}(1/\epsilon)\}$.

3. Based on the first term $\frac{L_f(f(x_0) - f(x^*))}{\eta SK} \leq \frac{1}{2}\epsilon$, the total number of iteration is $T = SK = \frac{2L_f(f(x_0) - f(x^*))}{\eta\epsilon}$.

Thus, based on the above parameters' setting, we can ensure that $\mathbb{E}[\|\nabla f(\hat{x}_k^s)\|^2] \leq \epsilon$.

Based on the parameters' setting, that is $D = \min\{n, \mathcal{O}(1/\epsilon)\}$, $A = \min\{n, \mathcal{O}(1/\eta)\}$, $K \leq \mathcal{O}(1/\eta^{3/2})$, and $T = \mathcal{O}(1/(\epsilon\eta))$, we have,

$$\mathcal{O}\left(\frac{T}{K}(D + KA)\right) = \mathcal{O}\left(\frac{1}{\epsilon\eta}\left(\frac{D}{K} + A\right)\right)$$

$$= \mathcal{O}\left(\frac{1}{\epsilon\eta}\left(\min\left\{n, \frac{1}{\epsilon}\right\}\eta^{3/2} + \frac{1}{\eta}\right)\right)$$

$$= \mathcal{O}\left(\frac{1}{\epsilon}\left(\min\left\{n, \frac{1}{\epsilon}\right\}\eta^{1/2} + \frac{1}{\eta^2}\right)\right)$$

$$\geq \mathcal{O}\left(\min\left\{\frac{1}{\epsilon^{9/5}}, \frac{n^{4/5}}{\epsilon}\right\}\right),$$

where the optimal $\eta = \min\{1/n^{2/5}, \epsilon^{2/5}\}$. □

## D    PROOF FOR THE MINI-BATCH OF THE SCCG TO THE COMPOSITION PROBLEM

We provide the Mini-batch version of SCCG:

The following lemma is distinguish with Lemma 5 in which the estimated gradient $\gamma$ is obtained through $b$ times repeat.

**Lemma 7.** *Suppose Assumption 1 holds, for $\hat{G}_k$ defined in (3.3) and $\Lambda$ defined in Algorithm 2 with $\mathcal{D} = [\mathcal{D}_1, \mathcal{D}_2]$ and $D = |\mathcal{D}_1| = |\mathcal{D}_2|$, we have*

$$\mathbb{E}_{i_k, j_k, \mathcal{A}, \mathcal{D}}\|\Lambda - \nabla f(x_k)\|^2 \leq 5B_G^4L_F^2\left(\frac{L_f^2}{bB_G^4L_F^2} + 4\frac{\mathbb{I}(A < n)}{A} + 4\frac{\mathbb{I}(D < n)}{D}\right)\mathbb{E}\|x_k - \tilde{x}_s\|^2$$

$$+ 20B_G^2L_F^2\frac{\mathbb{I}(D < n)}{D}H_1 + 5\frac{\mathbb{I}(D^2 < n^2)}{D^2}H_2,$$

*Proof.* Through adding and subtracting the term of $\frac{1}{b}\sum_{(i,j)\in I_b}(\partial G_j(x_k))^\mathsf{T}\nabla F_i(G(x_k))$,

$\frac{1}{b}\sum_{(i,j)\in I_b}(\partial G_i(\tilde{x}_s))^\mathsf{T}\nabla F_i(G(\tilde{x}_s))$, and $(\partial G(\tilde{x}_s))^\mathsf{T}\nabla F(G(\tilde{x}_s))$, $(\partial G_{\mathcal{D}_1}(\tilde{x}_s))^\mathsf{T}\nabla F_{\mathcal{D}_1}(G(\tilde{x}_s))$, we have

$\mathbb{E}\|\Lambda - \nabla f(x_k)\|^2$

---

**Algorithm 2** Mini-batch version of SCCG

**Require:** $K$, $S$, $\eta$ (learning rate), $\tilde{x}_0$ and $\mathcal{D} = [\mathcal{D}_1, \mathcal{D}_2]$
  **for** $s = 0, 1, 2, \cdots, S-1$ **do**
      Sample from $[n]$ for D times to form mini-batch $\mathcal{D}_1$
      Sample from $[n]$ for D times to form mini-batch $\mathcal{D}_2$
      $\nabla \hat{f}_{\mathcal{D}}(\tilde{x}_s) = (\partial G_{\mathcal{D}_1}(\tilde{x}_s))^{\mathsf{T}} \nabla F_{\mathcal{D}_2}(G_{\mathcal{D}_1}(\tilde{x}_s))$
      $x_0 = \tilde{x}_s$
      **for** $k = 0, 1, 2, \cdots, K-1$ **do**
         Sample from $[n]$ to form mini-batch $\mathcal{A}$
         $\hat{G}_k = G_{\mathcal{A}}(x_k) - G_{\mathcal{A}}(\tilde{x}_s) + G_{\mathcal{D}_1}(\tilde{x}_s)$
         $\Lambda_0 = 0$
         **for** t=1,...,b **do**
            Uniformly and randomly pick $i_k$ and $j_k$ from $[n]$
            Compute the estimated gradient $\nabla \tilde{f}_k$ from (3.4)
            $\Lambda_{t+1} = \Lambda_t + \nabla \tilde{f}_k$
         **end for**
         $\Lambda = \Lambda_b / b$
         $x_{k+1} = x_k - \eta \Lambda$
      **end for**
      Update $\tilde{x}_{s+1} = x_K$, or $\tilde{x}_{s+1} = x_r$, $r$ is randomly selected from $[K-1]$
  **end for**
  Output: $\hat{x}_k^s$ is uniformly and randomly chosen from $s \in \{0, ..., S-1\}$ and $k \in \{0, .., K-1\}$.

---

$$\overset{①}{\leq} 5\mathbb{E} \left\| \frac{1}{b} \sum_{(i,j) \in I_b} (\partial G_j(x_k))^{\mathsf{T}} \nabla F_i(G(x_k)) - (\partial G_j(\tilde{x}_s))^{\mathsf{T}} \nabla F_i(G(\tilde{x}_s)) - \left( \nabla f(x_k) - (\partial G(\tilde{x}_s))^{\mathsf{T}} \nabla F(G(\tilde{x}_s)) \right) \right\|^2$$

$$+ 5\mathbb{E} \left\| \frac{1}{b} \sum_{(i,j) \in I_b} (\partial G_j(x_k))^{\mathsf{T}} \nabla F_i(\hat{G}_k) - (\partial G_j(x_k))^{\mathsf{T}} \nabla F_i(G(x_k)) \right\|^2$$

$$+ 5\mathbb{E} \left\| \frac{1}{b} \sum_{(i,j) \in I_b} (\partial G_j(\tilde{x}_s))^{\mathsf{T}} \nabla F_i(G(\tilde{x}_s)) - (\partial G_j(\tilde{x}_s))^{\mathsf{T}} \nabla F_i(G_{\mathcal{D}_1}(\tilde{x}_s)) \right\|^2$$

$$+ 5\mathbb{E} \left\| \nabla \hat{f}_{\mathcal{D}}(\tilde{x}_s) - (\partial G_{\mathcal{D}_1}(\tilde{x}_s))^{\mathsf{T}} \nabla F_{\mathcal{D}_2}(G(\tilde{x}_s)) \right\|^2$$

$$+ 5\mathbb{E} \left\| (\partial G_{\mathcal{D}_1}(\tilde{x}_s))^{\mathsf{T}} \nabla F_{\mathcal{D}_2}(G(\tilde{x}_s)) - (\partial G(\tilde{x}_s))^{\mathsf{T}} \nabla F(G(\tilde{x}_s)) \right\|^2$$

$$\overset{②}{\leq} \frac{5}{b} L_f^2 \mathbb{E} \|x_k - \tilde{x}_s\|^2 + 5 B_G^2 L_F^2 \mathbb{E} \left\| \hat{G}_k - G(x_k) \right\|^2 + 5 B_G^2 L_F^2 \mathbb{E} \|G(\tilde{x}_s) - G_{\mathcal{D}_1}(\tilde{x}_s)\|^2$$

$$+ 5 B_G^2 L_F^2 \mathbb{E} \|G(\tilde{x}_s) - G_{\mathcal{D}_1}(\tilde{x}_s)\|^2 + 5 \frac{\mathbb{I}(D^2 < n^2)}{D^2} H_2$$

$$\overset{③}{\leq} 5 B_G^4 L_F^2 \left( \frac{L_f^2}{b B_G^4 L_F^2} + 4 \frac{\mathbb{I}(A < n)}{A} \right) \mathbb{E} \|x_k - \tilde{x}_s\|^2 + 40 B_G^2 L_F^2 \frac{\mathbb{I}(D < n)}{D} H_1 + 5 \frac{\mathbb{I}(D^2 < n^2)}{D^2} H_2,$$

where ① follows from $\|a_1 + a_2 + a_3 + a_4 + a_5\|^2 \leq 5a_1^2 + 5a_2^2 + 5a_3^2 + 5a_4^2 + 5a_5^2$, and ② is based on $\mathbb{E}[\|X - \mathbb{E}[X]\|^2] = \mathbb{E}[X^2 - \|\mathbb{E}[X]\|^2] \leq \mathbb{E}[X^2]$ and Lemma 1, the smoothness of $F_i$, the bounded Jacobian of $G(x)$ and the smoothness of $F$ in Assumption 1, and the upper bound of variance in Lemma 2. ③ is based on Lemma 3. $\qquad \square$

**Proof of Corollary 2**

*Proof.* Based on the parameters' setting, that is $D = \min\{n, \mathcal{O}(1/\epsilon)\}$, $A = \min\{n, \mathcal{O}(b/\eta)\}$, $K \leq \mathcal{O}\left(b^{1/2}/\eta^{3/2}\right)$, and $T = \mathcal{O}\left(1/(\epsilon\eta)\right)$, we have,

$$
\begin{aligned}
\mathcal{O}\left(\frac{T}{K}(D+KA)\right) &= O\left(\frac{\eta^{3/2}}{\epsilon b^{1/2}\eta}\left(\min\left\{n, \frac{1}{\epsilon}\right\} + \frac{b^{1/2}b}{\eta^{3/2}\eta}\right)\right) = O\left(\frac{\eta^{1/2}}{\epsilon b^{1/2}}\left(\min\left\{n, \frac{1}{\epsilon}\right\} + \frac{b^{3/2}}{\eta^{5/2}}\right)\right) \\
&= \frac{1}{\epsilon b^{1/2}}O\left(\min\left\{n, \frac{1}{\epsilon}\right\}\eta^{1/2} + \frac{b^{3/2}}{\eta^2}\right) \\
&\geq \frac{1}{b^{1/5}}O\left(\min\left\{\frac{n^{4/5}}{\epsilon}, \frac{1}{\epsilon^{9/5}}\right\}\right),
\end{aligned}
$$

where the optimal $\eta = b^{3/5}\min\left\{1/n^{2/5}, \epsilon^{2/5}\right\}$. $\qquad\square$

## E EXPERIMENT

### E.1 RISK-AVERSE LEARNING

To verify the effectiveness of the algorithm, we use the mean-variance optimization in portfolio management[7]:

$$
\min_{x\in\mathbb{R}^d} -\frac{1}{n}\sum_{i=1}^n \langle r_i, x\rangle + \frac{1}{n}\sum_{i=1}^n(\langle r_i, x\rangle - \frac{1}{n}\sum_{i=1}^n \langle r_i, x\rangle)^2,
$$

where $r_i \in \mathbb{R}^N, i \in [n]$ is the reward vector, and $x \in \mathbb{R}^N$ is the invested quantity. The objective function can be transformed as the composition of two finite-sum functions in (1.1) with the following forms:

$$
\begin{aligned}
G_j(x) &= [x, \langle r_j, x\rangle]^\mathsf{T}, \, y = \frac{1}{n}\sum_{j=1}^n G_j(x) = [y_1, y_2]^\mathsf{T}, \\
F_i(y) &= -\langle r_i, y_1\rangle + (\langle r_i, y_1\rangle - y_2)^2, j, i \in [n].
\end{aligned}
$$

where $y_1 \in \mathbb{R}^M$ and $y_2 \in \mathbb{R}$.

Note that the function $G_j(x) = [x, \langle r_j, x\rangle]$, and the corresponding Jacobian is $[I, e]^\top$, where $I \in \mathbb{R}^{N\times N}$ is a unit matrix, and $e \in \mathbb{R}^{N\times 1}$ is all-ones vector. It is straightforward to prove that the norm of the Jacobian is bounded, i.e. $G_j(x)$ is $B_G$-Lipschitz. We choose such example of the composition problem to verify the efficiency of the proposed algorithms, because it has been widely used in related researches (Lian et al., 2017; Wang et al., 2017; Lin et al., 2018). The source code package will be released as soon as possible to ensure the reproducibility.

### E.2 NON-LINEAR EMBEDDING

For the non-convex function, we apply the proposed SCCG method to the nonlinear embedding problem in (1.2). We consider the distance of low-dimension space between $x_i$ and $x_j$ as $1/(1 + \|x_i - x_j\|^2)$, $i, j \in [n]$. Then, the problem can be formulated as the problem in (1.1). In particular,

$$
\frac{1}{n}\sum_{i=1}^n F_i(y) = \frac{1}{n}\sum_{i=1}^n F_i\left(\frac{1}{n}\sum_{j=1}^n G_j(x)\right),
$$

where

$$
y = \frac{1}{n}\sum_{j=1}^n G_j(x);
$$

$$
G_j(x) = \left[x, \frac{n}{1 + \|x_1 - x_j\|^2} - 1, ..., \frac{n}{1 + \|x_n - x_j\|^2} - 1\right]^\mathsf{T};
$$

$$
F_i(y) = n\sum_{k=1}^n p_{k|i}(\|y_i - y_k\|^2 + \log(y_{n+k})), i, j \in [n].
$$

---

[7]This formulation is just used to verify our proposed algorithm.

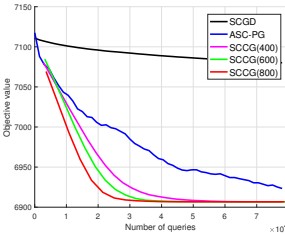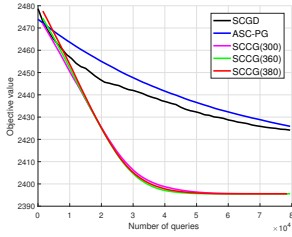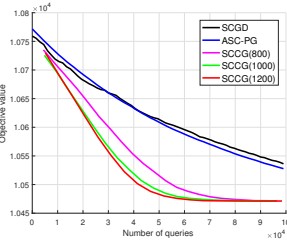

Figure 3: Non-convex: Comparison of the objective function between SCGD, ASC-PG and SCCG; Dataset (from left to right): mnist, olivettifaces and coil20.

Note that, consider the function $g(x) = \frac{1}{1+x^2}$, its gradient is $\nabla g(x) = \frac{2x}{(1+x^2)^2}$. For different value of $x$, we can see that

$$|x| \geq 1 \Rightarrow \left(1 + x^2\right)^2 \geq x;$$
$$|x| < 1 \Rightarrow \left(1 + x^2\right)^2 \geq 1 > x.$$

Thus, we obtain $|\nabla g(x)| \leq 2$, which is upper bounded. Based on this results, we can obtain that the norm of Jacobian is also bounded. Moreover, in practice, wo do not compute the Jacobian directly as the dimension is large. The matrix of Jacobian is sparse due to the random subset, which greatly save much space.

We consider three datasets: mnist, Olivetti faces and COIL-20 with different sample sizes and dimensions, $1000\times 784$, $400 \times 4096$ and $1440\times 16384$. Our experiment is to verify our proposed algorithm, thus, we set $\mathcal{D}_1 = \mathcal{D}_2$ in default and choose three different sizes of sample set $\mathcal{D}_1$, which are smaller than $n$. For example, for the case of mnist, we choose $|\mathcal{D}_1| = 400, 600, 800$, which are denoted as SCCG (400), SCCG (600) and SCCG (800). Furthermore, we also set $|\mathcal{A}|\approx n^{2/3}$, where $n$ is the total number of samples. Figure 2(in the main paper) shows the norm of the gradient, and Figure 3 shows the function value. We compare our algorithm with stochastic gradient based method (SCGD and ASC-PG) and observe that our proposed algorithm is better than SCGD and ASC-PG on both the norm of the gradient and objective function.

### E.3 REINFORCEMENT LEARNING

We consider the policy value evaluation in reinforcement learning. Let the policy of interest be $\pi$, total states be $S$, and the value function of state be $V^\pi$ at state $s_1$,

$$V^\pi(s_1) = \mathbb{E}_\pi \{R_{s_1,s_2} + \gamma V^\pi(s_2)|s_2\}, s_1, s_2 \in [S],$$

where $R_{s_1,s_2}$ is the reward of moving from state $s_1$ to $s_2$, and the expectation is taking over state $s_2$ conditioned on state $s_1$. We assume $V^\pi(s) \approx \Phi_s^T w^*$ for some $w^* \in R^d$, where $\Phi$ is the linear map of the feature used to approximate the value of the state. Then, the problem can be formulated as the Bellman residual minimization problem, that is

$$\min_w \sum_{i=1}^S \left( \langle \Phi_i, w \rangle - \sum_{j=1}^S P_{i,j}^\pi \left( R_{i,j} + \gamma \langle \Phi_j, w \rangle \right) \right)^2,$$

where $\gamma$ is a discount factor, $r_{ij}$ is the random reward of transition from $i$ to state $j$. Our proposed algorithm can be applied to the above problem, which can be formulated as the composition problem by taking

$$g_j(w) = S\left[\langle \Phi_1, w \rangle, ..., \langle \Phi_2, w \rangle, P_{1,j}^\pi \left( R_{1,j} + \gamma \langle \Phi_j, w \rangle \right), ..., P_{S,j}^\pi \left( R_{S,j} + \gamma \langle \Phi_j, w \rangle \right)\right]^T;$$
$$g(w) = \sum_{j=1}^S g_i(w) = y = \begin{bmatrix} y_1 \\ y_2 \end{bmatrix};$$
$$f_i(y) = S\|y_{1,i} - y_{2,i}\|^2.$$

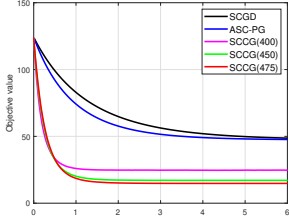 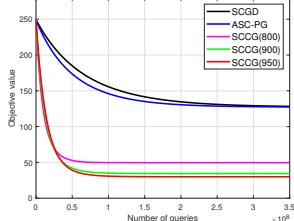 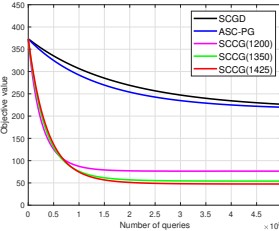

Figure 4: Reinforcement Learning application: Comparison of the objectively values between SCGD, ASC-PG and SCCG (including three different values of $|\mathcal{D}_1| = 0.95 * n, 0.9 * n, 0.8n$); Dataset (from left to right): n=500,1000, and 1500.

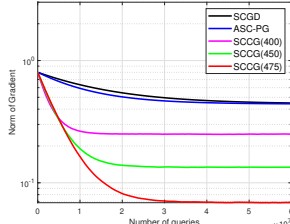 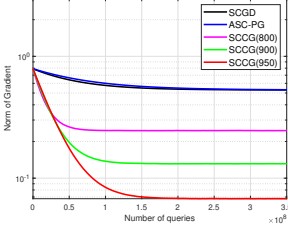 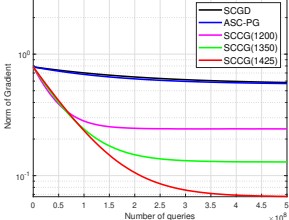

Figure 5: Reinforcement Learning application: Comparison of the norm of gradient between SCGD, ASC-PG and SCCG (including three different values of $|\mathcal{D}_1| = 0.95 * n, 0.9 * n, 0.8n$); Dataset (from left to right): n=500,1000, and 1500.

In the experiments, parameters $P^\pi$, $\Phi$ and $R$ are randomly selected. We implement on three data with the size of $n = 500, 1000, 1500$. And we set $|\mathcal{D}_1| = 0.95 * n, 0.9 * n, 0.8n$, respectively for different value of $n$. We set $b = |\mathcal{A}| \approx n^{2/3}$ based on our theory analysis. Figure 4 and 5 show the experimental results, which demonstrate that our proposed method is better than the non-variance reduction based methods SCGD and ASC-PG on both the objective value and the norm of the gradient.

