# OpenReview forum: "Stochastically Controlled Compositional Gradient for the Composition problem"
_ICLR.cc/2020/Conference — Reject_

### Official Review · AnonReviewer3 · 2019-10-22
**Official Blind Review #3**

**Rating:** 3

**Review:**

In the paper, the authors consider composition problems and use the stochastically controlled stochastic gradient method (SCSG) to approximate the gradient G(x) and \nabla f(x). The authors also provide convergence analysis of the proposed method for strongly convex problems and non-convex problems. Authors then conduct experiments on the
mean-variance optimization in portfolio management task and the nonlinear embedding problem, results show that the proposed method is faster.

The following are my concerns:
1) There are several important related works missing in the paper, e.g., [1][2].
2) The convergence results of the proposed method in the paper are not state-of-the-art. For a strongly convex case, the result in the paper is O( n+ k^2 min(n, 1/u^2) log1/e), it is not necessarily better than O(n+k^3 log 1/e) in [1] or O(n+kn^{2/3} log 1/e). For a non-convex case, the result in the paper is O(\min{1/e^{9/5}, n^{4/5} / e}), it  is not necessarily better than O(n^{2/3}/e) in [1] or [2].
3) More compared results should be conducted in the experiments, e.g. [1][2].

[1]Huo, Zhouyuan, et al. "Accelerated method for stochastic composition optimization with nonsmooth regularization." Thirty-Second AAAI Conference on Artificial Intelligence. 2018.
[2] Zhang, Junyu, and Lin Xiao. "A Composite Randomized Incremental Gradient Method." International Conference on Machine Learning. 2019.



**Experience Assessment:**

I have published in this field for several years.

**Review Assessment: Checking Correctness Of Derivations And Theory:**

I assessed the sensibility of the derivations and theory.

**Review Assessment: Checking Correctness Of Experiments:**

I carefully checked the experiments.

**Review Assessment: Thoroughness In Paper Reading:**

I read the paper at least twice and used my best judgement in assessing the paper.

---

> ### Author Response · Authors · 2019-11-08
> **Response to Reviewer 3**
>
> Thanks for your suggestions.
> We will give more discussion of the query complexity (QC) to demonstrate that our proposed method is more general.  Comparing with [1] and [2], query complexity of our proposed method is better than [1] and [2]  when the number of n is large. The QC of [1] and [2] are the mini-batch version, while we separately analyzed the non-mini-batch and mini-batch. The updated revised paper clearly shows the comparison of mini-batch versions among different algorithms, and also shows that our proposed method is better than [1] and [2] when n is large.
>
> For non-convex:
> 1)	When $n<\frac{1}{\epsilon}$, the QC of our proposed method is non-mini-batch $\mathcal{O}(n^{4/5} / \epsilon)$ and mini-batch version $\mathcal{O}(n^{4/5} /( b^{1/5} \epsilon))$, where b is the mini-batch size. QC of [1] and [2]  are equal to our mini-batch version $\mathcal{O}(n^{4/5} /( b^{1/5} \epsilon))$, when $b=n^{2/3}$. Because $\mathcal{O}(n^{4/5} /( b^{1/5} \epsilon))= \mathcal{O}(n^{2/3} /\epsilon)$. Furthermore, $b\le min\{n,1/\epsilon\}^{2/3}$ is based on the requirement of $\eta <1$, which is clearly updated in the revised paper.
> 2)	When $n >\frac{1}{\epsilon}$, this is our key motivation for our proposed method. QC of our proposed methods is better than [1] and [2]. That is $b= 1/\epsilon ^{2/3}$, the QC of our algorithm is $\mathcal{O}(1/\epsilon ^{5/3})$, which is better than [1][2]. Furthermore, when n is large enough, it is not proper to compute the gradient at each epoch, or when $1/\epsilon$ is not large enough, we can quickly get the desired results.
>
> For strongly convex:
> mini-batch QC of our proposed algorithm is $\mathcal{O}(min\{n,1/\epsilon \mu^2\}+ 1/\mu$ $ min\{n, 1/\mu^2\} log(1/\epsilon))$ by setting $b=\frac{1}{\mu}$(from the stepsize $\eta<1$), which  is better than [1] and [2] when n is large. This is the motivation of our proposed method when facing number of $n$.
>
> In order to verify our proposed method, we also add another reinforcement learning application to our proposed method. The experimental results are shown in the appendix. We compare our proposed algorithm with compositional gradient methods, which show that the performance of our proposed is better on both objective value and the norm of the gradient.

---

### Official Review · AnonReviewer1 · 2019-10-23
**Official Blind Review #1**

**Rating:** 6

**Review:**

This paper proposes a new method for empirical composition problems to which the vanilla SGD is not applicable because it has a finite-sum structure inside non-linear loss functions. A proposed method (named SCCG) is a combination of stochastic compositional gradient descent (SCGD) and stochastically controlled stochastic gradient (SCSG). In a theoretical analysis part, a linear convergence rate and a sub-linear convergence rate are derived under the strong convex and non-convex settings, respectively. In experiments, the superior performance of the method to competitors is verified on both strongly convex and non-convex problems.

Clarity:
The paper is clear and well written.

Quality:
The work is of good quality and is technically sound.

Significance:
The problem treated in this paper is important and contains several applications as mentioned in the paper. Hence, developing an efficient method for this problem is important and interesting. Although derived convergence rates are better than existing primal methods, this paper lacks a comparison with the recently proposed primal-dual method by [A.Devraj & J.Chen (2019)].

[A.Devraj & J.Chen (2019)] Stochastic Variance Reduced Primal Dual Algorithms for Empirical Composition Optimization. NeurIPS, 2019.

A convergence rate obtained in [A.Devraj & J.Chen (2019)] seems faster than that of SCCG for ill-conditioned strongly convex problems. However, there exists a certain setting (large-scale setting) where SCCG outperforms their method. Thus, the contribution of the paper is not lost, but it is better to compare SCCG with the method in [A.Devraj & J.Chen (2019)], empirically and theoretically.
If the authors can show an empirical advantage over their method, it will make the paper stronger.

-----
Update:
I thank the authors for the response and hard work. I am convinced of the advantage of the proposed method. I would like to keep my score.

**Experience Assessment:**

I have read many papers in this area.

**Review Assessment: Checking Correctness Of Derivations And Theory:**

I assessed the sensibility of the derivations and theory.

**Review Assessment: Checking Correctness Of Experiments:**

I did not assess the experiments.

**Review Assessment: Thoroughness In Paper Reading:**

I made a quick assessment of this paper.

---

> ### Author Response · Authors · 2019-11-08
> **Response to Reviewer 1**
>
> Thanks for your suggestions. We will cite this paper to make more discussion.  [A.Devraj & J.Chen (2019) ] consider the strongly convex composition problem based on the primal-dual method. Our query complexity is better than [A.Devraj & J.Chen (2019) ] when n is large. This is because our estimated gradient depends on the relationship between $n$ and $1/\epsilon$, does not depend only on $n$. Thus the query complexity is more general than the previous. The target of our experiment is to compare our proposed method to the non-variance reduction based method, in which the complexity does not contain $n$ . What’s more, besides the two applications, we also add a reinforcement learning application to our proposed algorithm, which also demonstrate that our proposed method is better than the compositional gradient methods. The added experimental results are shown in the appendix.

---

### Official Review · AnonReviewer2 · 2019-10-31
**Official Blind Review #2**

**Rating:** 3

**Review:**

The paper proposes a variance reduction based algorithm to solve compositional problems. The idea comes from the stochastically controlled stochastic gradient (SCSG) methods. The paper applies the idea from SCSG to estimating the inner function G(x) and the gradient \nabla f_k to solve compositional problems. The paper provides a theoretical analysis of the query complexity of the algorithm in both convex and non-convex setting. The experiments show the performance of the proposed algorithm is better than other recent methods. The paper seems to be the first attempt to extending stochastically controlled functions to the compositional problems. However, I vote for rejecting this submission for the following concerns. (1) Since SCSG is a member of the SVRG family of algorithms, the difference between this paper and [Xiangru Lian, Mengdi Wang, and Ji Liu, 2017] is not significant enough, especially in the algorithm design and the proof of the theoretical theorem. (2) The formulation of the compositional problems comes from reinforcement learning, risk-averse learning, nonlinear embedding, etc. However, the experiments are only performed on nonlinear-embedding problems. I think performing the experiments on different kinds of problems will be helpful to justify the significance.

**Experience Assessment:**

I have read many papers in this area.

**Review Assessment: Checking Correctness Of Derivations And Theory:**

I carefully checked the derivations and theory.

**Review Assessment: Checking Correctness Of Experiments:**

I assessed the sensibility of the experiments.

**Review Assessment: Thoroughness In Paper Reading:**

I read the paper thoroughly.

---

> ### Author Response · Authors · 2019-11-08
> **Response to Reviewer 2**
>
> Thanks for your suggestions.
> Although both the proposed stochastically controlled compositional gradient (SCCG) and C-SVRG [Xiangru Lian, Mengdi Wang, and Ji Liu, 2017] are variance reduction-based methods, we respectively argue they are significantly different. This is mainly because:
>         (a) The estimated gradient involves the inner function estimation and biased gradient estimation, which is greatly different from C-SVRG. The motivation for the proposed estimated gradient is that we can avoid the direct computation of inner function and the gradient at each epoch when $n$ is large, which is our advantage over C-SVRG.
>         (b) The proposed estimated gradient makes the challenge for the proof as the estimated inner function is also biased. However, we give a new proof for the convergence, which is essentially different from C-SVRG. What’s more, our proposed estimated gradient included two kinds of stochastically controlled gradients, which is more general than C-SVRG.
>         (c) The query complexity of our proposed method is better than C-SVRG when $n$ is large.
>
> Our experiments include two applications: nonlinear-embedding problems for non-convex and risk-averse learning for strongly convex, which are both used to verify our proposed algorithms.  Moreover, we also add the reinforcement learning application to our proposed algorithms, which also demonstrate that our proposed algorithm is better than the non-variance-reduction based method. The corresponding experimental results are in the appendix.

---

### Decision · Program_Chairs · 2019-12-19

**Decision:**

Reject

**Comment:**

All the reivewers find the similarity between this paper and the references in terms of the algorithm and the proof. The theoretical results may not better than the existing results.